# Temperature overshoot responses to ambitious forestation in an Earth System Model

Yiannis Moustakis [1] ✉, Tobias Nützel [1], Hao-Wei Wey[2], Wenkai Bao[1] & Julia Pongratz[1,3]

Despite the increasing relevance of temperature overshoot and the rather ambitious country pledges on Afforestation/Reforestation globally, the mitigation potential and the Earth system responses to large-scale non-idealized Afforestation/Reforestation patterns under a high overshoot scenario remain elusive. Here, we develop an ambitious Afforestation/Reforestation scenario by harnessing 1259 Integrated Assessment Model scenarios, restoration potential maps, and biodiversity constraints, reaching 595 Mha by 2060 and 935 Mha by 2100. We then force the Max Planck Institute's Earth System Model with this scenario which yields a reduction of peak temperature by 0.08 °C, end-of-century temperature by 0.2 °C, and overshoot duration by 13 years. Afforestation/Reforestation in the range of country pledges globally could thus constitute a useful mitigation tool in overshoot scenarios in addition to fossil fuel emission reductions, but socio-ecological implications need to be scrutinized to avoid severe side effects.

Reaching net-zero greenhouse gas emissions as required by the Paris Agreement's goals will require the deployment of Carbon Dioxide Removal (CDR) methods, compensating at least for hard-to-abate emissions[1–3]. Pursuing a more ambitious 1.5 °C (compared to pre-industrial levels) goal implies a more limited remaining emission budget, and thus rapid decarbonization, major societal transformations, and large-scale CDR application[1,4]. Thus, CDR deployment has been at the spotlight of scientific research recently[5,6]. Future scenarios generated by Integrated Assessment Models (IAMs) that keep to 1.5 °C by 2100 typically exhibit excess positive emissions early on which are then sharply reduced, leading to a temperature overshoot (i.e., period during which mean global temperature exceeds intended warming level, here 1.5 °C, before returning to it)[7,8]. This delay of early action in IAMs emerges due to the high upfront investments and near-term mitigation costs required to limit the overshoot[9,10], but caution is needed since the normalization of the overshoot idea could possibly facilitate a political flexibility associated with a lack of action[11]. Overshoot is becoming increasingly more relevant due to incredible country pledges[12,13], and concerns of political feasibility and delay of

the required climate action[14,15]. Despite that, uncertainty still remains with regards to overshoot dynamics such as the peak temperature and overshoot duration and the associated Earth system responses and adaptation needs, which should thus be further studied[9,16].

Afforestation/Reforestation (AR), which has been practiced for decades, can be a useful mitigation tool[17] and combined with forest management practices constitutes virtually all (99.9%) of currently applied CDR[18]. In the literature a wide range of possible AR sequestration potentials has been reported, varying depending on the model used, land availability, $CO_2$ levels, and assumptions regarding the capacity of different biomes in sequestering $CO_2$ under a changing climate[7,19,20]. Apart from the potential of durably sequestering carbon (C) and mitigating warming, uncertainty remains regarding the permanence of C storage[21], possible impacts of large-scale AR on local climates[22], ecosystem biodiversity[19], food security[23], and the associated societal risks[7]. Still, recently the Land-Gap Report[24] has estimated that long-term and net-zero country-level mitigation pledges amount to quite ambitious land-based CDR commitments, which would require 633 Mha of AR globally by 2060, in addition to 551 Mha for the

[1]Ludwig-Maximilians-Universität in Munich, Munich, Germany. [2]GEOMAR Helmholtz Centre for Ocean Research Kiel, Kiel, Germany. [3]Max Planck Institute for Meteorology, Hamburg, Germany. ✉e-mail: yiannis.moustakis@geographie.uni-muenchen.de

restoration of degraded ecosystems. This estimation has recently been updated reaching likely up to 490 Mha of AR and up to 570 Mha of restoration[25]. Even though such levels of AR might seem rather high and their feasibility and associated socioeconomic risks have been questioned[24], it is only through decisive action and ambitious policies that we can diverge from the high-risk climate path that we are currently following[13]. Hence, the mitigation potential and impacts of large-scale AR in the range of country pledges should be further assessed.

The complexity of assessing ambitious AR deployment emerges since both the changes in atmospheric $CO_2$ following CDR application and the biogeophysical effects via altered moisture and surface energy fluxes can strongly affect the Earth system[26]. To capture these effects and feedbacks and thus get more reliable estimates of AR potentials and impacts, fully coupled Earth System Models (ESMs) need to be employed, which solve for the exchange of mass and energy between the land surface, the atmosphere and the ocean[27]. The assessment of biogeophysical effects has mostly been done for land use change, land management, and related land-cover changes (short "land use" in the following) in general. Studies investigated local or remote temperature[26,28], precipitation[29], and circulation changes[30], caused by changes in surface roughness and albedo[31], moisture recycling and net radiation[22], as well as their interplay with the biogeochemical effect (cooling) of the associated decrease in atmospheric $CO_2$[32].

Even though significant progress has been made in isolating and investigating such effects, studies in many cases employ fully idealized land use scenarios including massive (de)forestation globally, and cannot be thus used to confidently assess the effects of more realistic patterns[33–35]. This is especially the case with the multitude of results reported by deforestation studies, since it has recently been demonstrated that the responses of local climate to forest loss and gain are asymmetric[36]. Modeling efforts with a focus on assessing the impacts and mitigation potentials of more realistic AR scenarios have been rather limited. Some have been based on the available land use change scenarios associated with the Representative Concentration Pathway 4.5 (RCP4.5) from the Coupled Model Intercomparison Project 5 (CMIP5)[37–39]. In RCP4.5, however, the total ~800Mha forest area increase by 2100 also includes a forest expansion that the models simulate in response to climate change, and typically, by experimental design (comparison is only possible between various land use scenarios), AR can be assessed only together with land transitions other than AR, including avoided deforestation, and thus it is not possible to isolate the effect of CDR[39]. Recently, Matthews et al.[40] examined the mitigation potential of forest regrowth to 1920 levels by 2056, followed by a release of sequestered carbon during the second half of the century, and demonstrated that high AR levels, even when of temporary nature, can mitigate global temperature increase under a low overshoot scenario. However, Matthews et al.[40] employed an intermediate complexity climate model, which does not fully represent all the relevant processes, and an idealized spatiotemporal AR pattern. In the ongoing Land Use Model Intercomparison Project (LUMIP)[41] the land use pattern of the Shared Socioeconomic Pathway (SSP) SSP1-2.6 is considered. Even though this was intended to include substantial AR based on the marker IAM (IMAGE) scenario[42], translation to ESMs has failed to reproduce that signal yielding moderate AR levels[43], thus falling behind the ambitious amounts suggested by country pledges globally.

Overall, uncertainty still remains regarding the Earth system responses under temperature overshoot pathways[16]. Reported results show a significant variability of responses among ESMs, whose statistical significance, as well as the signal emergence from internal natural variability have not been robustly assessed, due to the lack of multiple (within and across model) ensemble members[16,44]. Yet, in most AR studies high future emission scenarios are used[34,39]. Importantly, such approaches suffer from the paradox of applying ambitious AR -implying strong coordinated climate action-, while following high-end emission scenarios -implying delayed and insufficient climate action. The normalization of the idea of coupling largely unmitigated warming scenarios with ambitious AR could connotate the establishment of AR as an alternative to emission reductions, and needs to be treated with caution[45]. From a biophysical perspective, high-end scenarios induce an enhanced $CO_2$ fertilization effect, thus strengthening vegetation carbon uptake, land surface fluxes, mitigation potentials and biogeochemical cooling[34,38,39,41]. Typically, they also yield intensified disturbances (e.g., wildfires) which could compromise the permanence of carbon stored in vegetation[46]. Evidently, the role of AR under scenarios with stronger emission reductions yielding a weaker vegetation fertilization, and hence less biogeochemical cooling, as well as lower disturbance rates, thus suffering less from CDR permanence overestimation, remains unclear.

To fill this gap, we investigate to what extent an ambitious constrained AR scenario in the range of total country pledges globally can mitigate a temperature overshoot emission scenario and alter its dynamics (duration, peak and end-of-century temperature), and whether biogeochemically-induced cooling can emerge from internal variability and compensate for any possible AR-induced biogeophysical warming. We first develop an AR scenario in the range of country pledges globally reaching 595Mha of AR in 2060 and 935 Mha by 2100, and then run fully coupled simulations with the Max Planck Institute's Earth System Model (MPI-ESM). Simulations are performed under a high-overshoot SSP5-3.4os emissions scenario which follows SSP5-8.5 until 2040, followed by rapid decarbonization, and reaching net-zero around 2070 and net-negative thereafter. To take into account the technoeconomic considerations, biogeophysical potentials, and biodiversity concerns, we harvest the recently published AR6 Scenarios Database (AR6-SD)[47], restoration potential maps, and maps of human influence. Beyond these considerations we also discuss in detail the possible socioeconomic barriers and tradeoffs of such ambitious AR implementation.

Our results show that ambitious AR in the range of country pledges globally can mitigate a temperature overshoot scenario by reducing the peak and end-of-century temperature by 0.08 °C and 0.2 °C respectively, as well as the duration of the overshoot by 13 years. Overall, biogeochemical cooling due to the reduction of $CO_2$ dominates, and at least compensates for any biogeophysically-induced warming.

## Results
### Afforestation/Reforestation pattern

Following AR6-SD estimates (Fig. 1, "Methods"), AR is applied at the expense of grazing land globally, while cropland areas and natural grassy and shrubby biomes remain unchanged. By considering restoration potential maps and biodiversity proxies (Fig. 2, "Methods"), the resulting AR scenario (Fig. 3) exhibits a strong increase in forest area over the 21st century, with slightly higher dynamics in the first half, reaching 595 (935) Mha of forest area increase by 2060 (2100) over current (2015) levels. This occurs at the expense of heavily managed pastures, which decrease by 472 (575) Mha and more lightly managed rangelands, which decrease by 123 (360) Mha. Latin America (256 Mha in 2100) and Africa (215 Mha) emerge as hotspots of forest area increase, followed by the Organization for Economic Cooperation and Development (OECD90) developed countries (201 Mha), Asia (162 Mha), and Eastern European countries and the reforming economies of the former Soviet Union (101 Mha). At the country level, the United States (142 Mha), Brazil (113 Mha), China (110 Mha), Russian Federation (64 Mha), and Argentina (24 Mha) carry out in total 48% of AR by 2100 (Supplementary Fig. 1). The scenario includes reforesting 514 Mha in 2100 of previously deforested areas globally, corresponding to 60% of total historical deforestation (851 Mha during 1850–2014 in MPI-ESM following the Land Use Harmonization 2[48] historical pattern) (Supplementary Fig. 2). The remaining 421 Mha are afforestation, which is employed by priority over the less biodiverse rangelands (Fig. 2 & Supplementary Fig. 2).

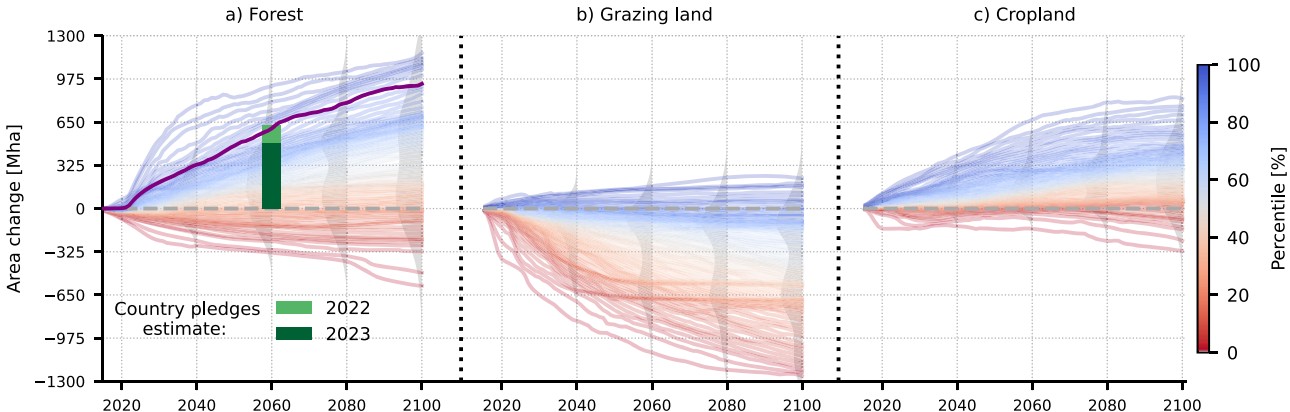

**Fig. 1 | Percentiles of the AR6 scenario database.** The figures present the different percentiles of global (**a**) forest cover, (**b**) grazing land, and (**c**) cropland change in Mha compared to 2015. Each line does not correspond to a specific scenario, but rather represents a different percentile of area change for each year, estimated by pooling all AR6 Scenario Database scenarios together. Highlighted is the 90th percentile for forest area change and its complementary threshold, the 10th percentile for cropland and grazing land change. The probability density plots for 2040, 2060, 2080, and 2100 indicative of the spread of the area change distribution are also shown in light gray. The country pledges global estimate presented in the Land-Gap report[24], and its 2023 update[25] are shown with the green bar.

**Fig. 2 | Overview of the AR scenario development.** Given that total rangelands (**a**) can have different levels of management and grazing intensity and hence biodiversity richness, we use the Very Low Human Influence (VLHI) map based on Riggio et al.[127] (**b**) to filter out grazing land that can be considered closer to a pristine state. Combined with pasture, the remaining available grazing land (**c**) constitutes the total grazing land that is considered for AR (**d**) in our framework. Available rangeland is further categorized into 4 biodiversity groups (**e**), based on the Low Human Influence (LHI) map based on Riggio et al.[127] (**f**). The Griscom et al.[19] restoration potential map (GRS, (**g**)) and Atlas of Forest and Landscape Restoration Opportunities[126] (ATL, (**h**)) are further used to guide the spatial pattern of AR, given an annual AR target for each world region. Allocation of the AR target across the gridcells is performed using the iterative process presented in the "Methods" section. The resulting AR pattern by 2100 for the target set in this study is presented in (**i**).

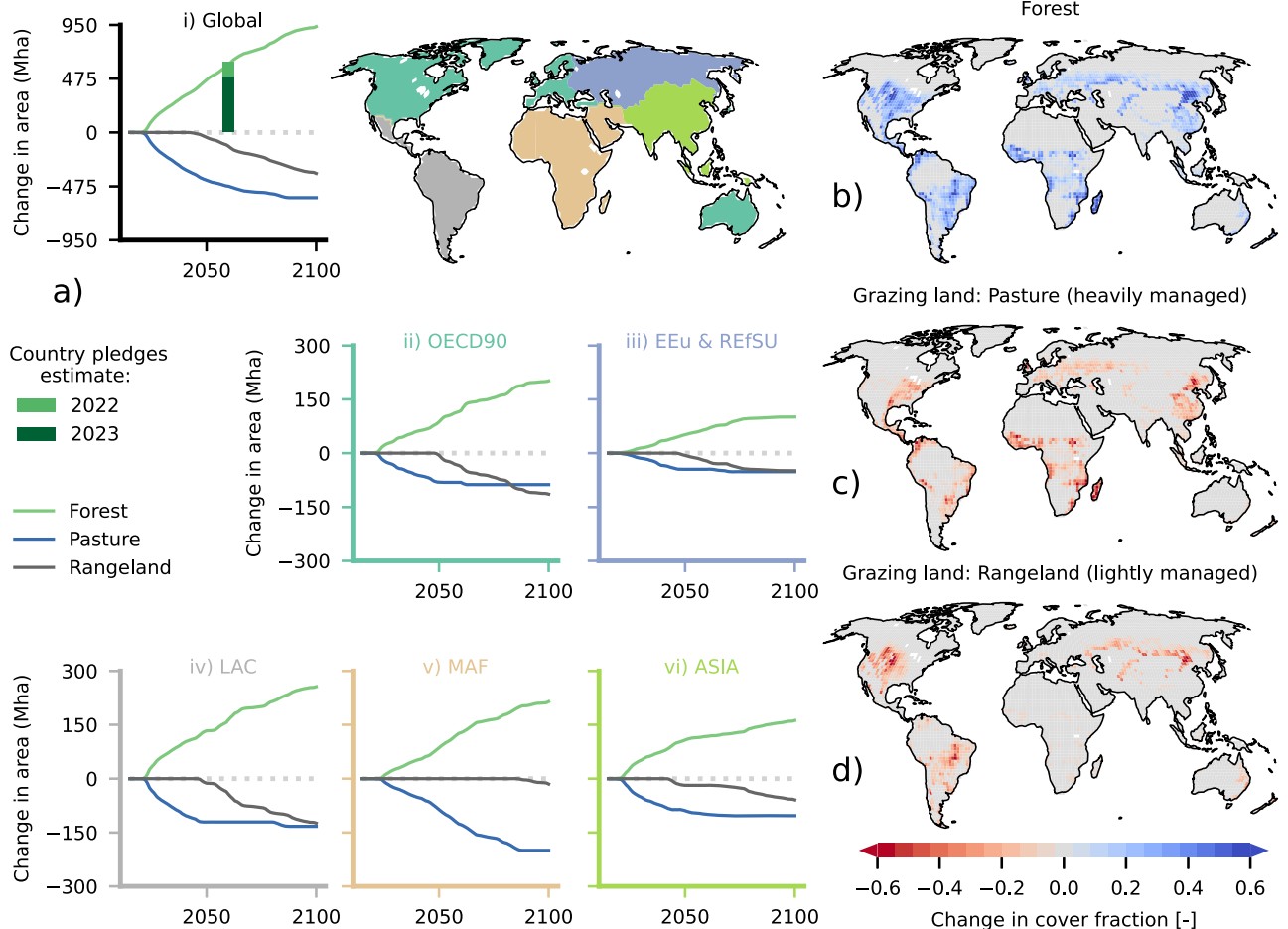

**Fig. 3 | Results of the AR scenario development (see "Methods" and Fig. 2).** The panels in (**a**) show the timeseries of forest area increase (in Mha) and how AR is allocated between pasture and rangelands i) globally and at the regional level for ii) Organization for Economic Cooperation and Development (OECD90) developed countries, (iii) Eastern Europe and reforming economies of the former Soviet Union (EEu & RefSU), (iv) Latin America (LAC), (v) Middle East and Africa (MAF), and (vi) Asia (the 5 world economic regions typically used in IAMs). The country pledges global estimate presented in the Land-Gap report[24], and its 2023 update[25] are also shown with the green bar in (i). In the right-column maps the resulting change in cover fraction of (**b**) forest cover (corresponding to Fig. 2i), (**c**) heavily managed grazing land (Pasture), and (**d**) lightly managed grazing land (Rangeland) from 2015 to 2100 is shown (other vegetation/land use including croplands remain unaltered).

## Mitigation of global temperature

The average global 2 m air temperature (compared to the 1850–1900 MPI-ESM average) and the global carbon fluxes are shown in Fig. 4. Under the reference (REF) simulation (constant land use at 2015 levels, see "Methods"), temperature overshoot lasts for ~65 ± 2 years (mean ± standard deviation estimated from smoothing and bootstrapping—see "Methods"), from 2035 up to 2099. The probability that end-of-century temperature will be within 1.5 °C over pre-industrial levels is 46%, with an average temperature of 1.51 °C (minimum–maximum of 5-year period from ensemble members: 1.27–1.87 °C—see "Methods"). In year 2058 ± 2, a peak temperature of 2.06 °C (1.89–2.29 °C) is reached, which follows the year during which atmospheric carbon reaches its maximum value (~ + 980 GtCO₂, 126 ppm in 2057 compared to 2015). Under the AR simulation, temperature overshoot lasts for ~52 ± 2 years, from 2036 up to 2087. The probability that the 1.5 °C target is returned to by the end of the century is 90%, with an average temperature of 1.31 °C (1.02–1.59 °C). In year 2058 ± 2, a peak temperature of 1.98 °C (1.8–2.23 °C) is reached, staying below the Paris Agreement 2 °C target with a probability of 48%. Peak temperature lags ~3 years after the maximum in atmospheric carbon. Comparing the REF and AR simulations suggests that the isolated effect of AR on average global temperature yields peak temperature reduced by ~0.08 °C, overshoot duration by ~13 years, and end-of-century temperature by ~0.2 °C. The impact of AR on global

temperature emerges already in ~2052 ± 2 years, when the temperature difference with REF starts becoming statistically significant (Fig. 4a). At that point, AR has reached 495 Mha, atmospheric carbon is lower by -110 GtCO₂ (30 PgC, 14 ppm), and oceanic carbon by -22 GtCO₂ (6 PgC), due to a total increase of ~132 GtCO₂ (36 PgC) in the land carbon sink (Fig. 4b). The temperature trajectory under AR is smoother, rather reaching a plateau before it starts leveling off. By examining different percentiles of global average daily temperature a similar behavior is demonstrated (Supplementary Fig. 3).

## CO₂ removal

The land C stock divergence of AR from REF is notable within few decades and continues until the end of the century (Fig. 4b). Over gridcells where AR is applied the land sequesters in total ~395 GtCO₂ (108 PgC) more compared to REF by 2100, with less carbon being sequestered over the forested rangelands, which are typically more arid (Fig. 4c). This is the net result of ~420 GtCO₂ (115 PgC) sink enhancement over 912 Mha of our AR area, and a weakening of the land C sink by -25 GtCO₂ (7 PgC) over the remaining 23 Mha, mostly in the tropics (despite the forest area increase). Overall, -249 GtCO₂ (68 PgC) correspond to the partial reversal of historical deforestation. Over gridcells where AR is not applied, a total weakening of CO₂ uptake by -13 GtCO₂ (4 PgC) emerges, most prominently over the Amazon. Therefore, the net land sink increase compared to REF is ~382 GtCO₂

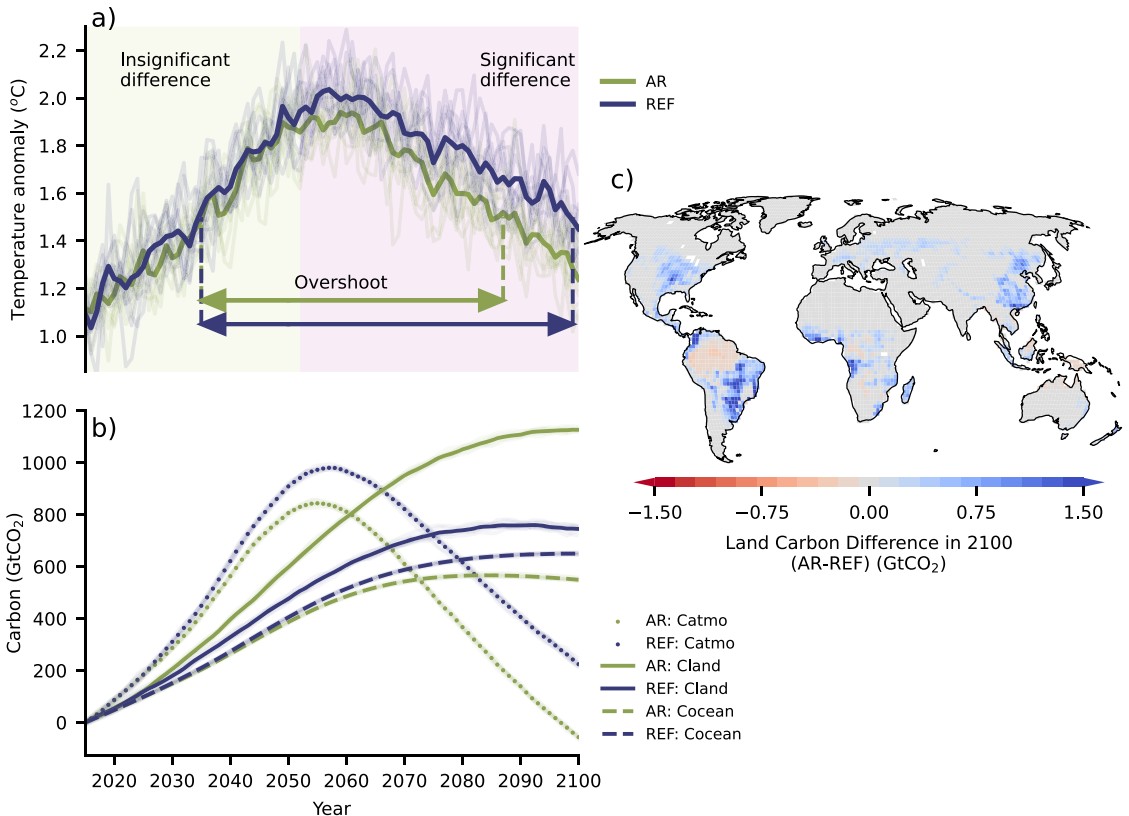

**Fig. 4 | Earth system responses. a** Average annual global 2 m air temperature (difference compared to pre-industrial era, expressed here as the 1850–1900 average) is plotted for each ensemble member (light lines) for both AR (green color) and REF (blue color) scenarios. The thick lines represent the ensemble mean (10 members) for each scenario. The colored arrows depict the overshoot duration for each scenario. The bright green shaded region indicates that there is no statistically significant difference between the two scenarios, while the purple shading suggests that a statistically significant difference exists (significance estimated as described in "Methods"). **b** The changes in global ensemble mean total atmospheric (Catmo), land (Cland), and oceanic (Cocean) carbon (GtCO$_2$) compared to 2015 are shown for each scenario. All ensemble members are plotted in light color, but overall variability is weak compared to the temperature variability. **c** Map showing the ensemble mean difference in carbon (GtCO$_2$ in each gridcell) stored on land between the AR and the REF scenario in 2100. Blue color indicates an increase in land carbon under AR.

(104 PgC) by 2100 in total, reducing the atmospheric CO$_2$ level by -281 GtCO$_2$ (77 PgC, 36 ppm), and the ocean sink by -101 GtCO$_2$ (27 PgC).

The rate of change in land C under both REF and AR mostly follows the signal of change in atmospheric C, which is driven by fossil fuel emissions (Supplementary Fig. 4). Under REF, the land C stock transits from being a sink to a source in -2090, whereas under AR the land C remains a sink, even though with weak and ever decreasing sequestration rates, becoming a source only in 2100. This suggests that under AR increased forest cover coupled with decreased CO$_2$ atmospheric levels compared to REF offer a -10-year buffer before the land C stock transits to a source behavior, allowing for more C to be sequestered during this period. However, the reverse image is observed when the ocean C sink is examined (Fig. 4 & Supplementary Fig. 4). Here, not only is ocean C uptake consistently weaker under AR compared to REF, due to the lower CO$_2$ concentration as a consequence of the CDR application, but in fact the ocean sink transits to a source behavior under AR in -2085. Under REF the ocean C sink does not transit to a source behavior, even though by 2100 the rate of increase is almost zero. This demonstrates that the ocean tends to partly offset the efficiency of applied CDR. On average, for every additional 100 GtCO$_2$ (27 PgC) sequestered in land under AR compared to REF 26 GtCO$_2$ (7 PgC) less carbon is taken up by the ocean, leading to a net removal of 74 GtCO$_2$ (20 PgC) from the atmosphere, which suggests a 74% efficiency.

Overall, the applied AR manages to sequester -4.49 GtCO$_2$/year more over land compared to REF on average, but eventually reduces

atmospheric CO$_2$ only by -3.32 GtCO$_2$/year on average, due to compensation by the ocean CO$_2$ fluxes. By 2100, this corresponds to a -41 GtCO$_2$ (11 PgC) gain over land compared to REF and -0.02 °C temperature reduction for every 100 Mha of forest area increase, which also suggests a -0.05 °C temperature reduction for every 100 GtCO$_2$ (27 PgC) additionally sequestered in land compared to REF. CDR efficiency expressed as temperature reduction per 100 GtCO$_2$ removed from the atmosphere compared to REF increases across time, reaching -0.07 °C in 2100 (Supplementary Fig. 5). Efficiency expressed in terms of C removal per 100 Mha increases until -2080 and then plateaus until the end of the century, reaching on average -30 GtCO$_2$ (8 PgC) (Supplementary Fig. 5).

AR sequestration rate defined as the rate of land C increase when AR and REF are compared shows a sharp increase to 3.9 GtCO$_2$/year in 2030 (27 GtCO$_2$ cumulatively), rising further to 6.3 GtCO$_2$/year in 2050 (121 GtCO$_2$) (Fig. 5). These high rates are sustained and strengthened until -2065 (7.1 GtCO$_2$/year, 222 GtCO$_2$), before they start levelling off, reaching 1.7 GtCO$_2$/year (382 GtCO$_2$) in 2100 even though at this point the individual land C stocks under REF and AR have both transited to a source behavior. When it comes to the rate of decrease in atmospheric CO$_2$ when AR and REF are compared, a removal rate of 3.39 GtCO$_2$/year is achieved by 2030 (24 GtCO$_2$), 5.3 GtCO$_2$/year by 2050 (103 GtCO$_2$), and 5.6 GtCO$_2$/year by 2065 (185 GtCO$_2$). By 2100, atmospheric CO$_2$ rates reach −0.21 GtCO$_2$/year (281 GtCO$_2$ in total), since additively the ocean and land C sources under AR are stronger than the land C source under REF.

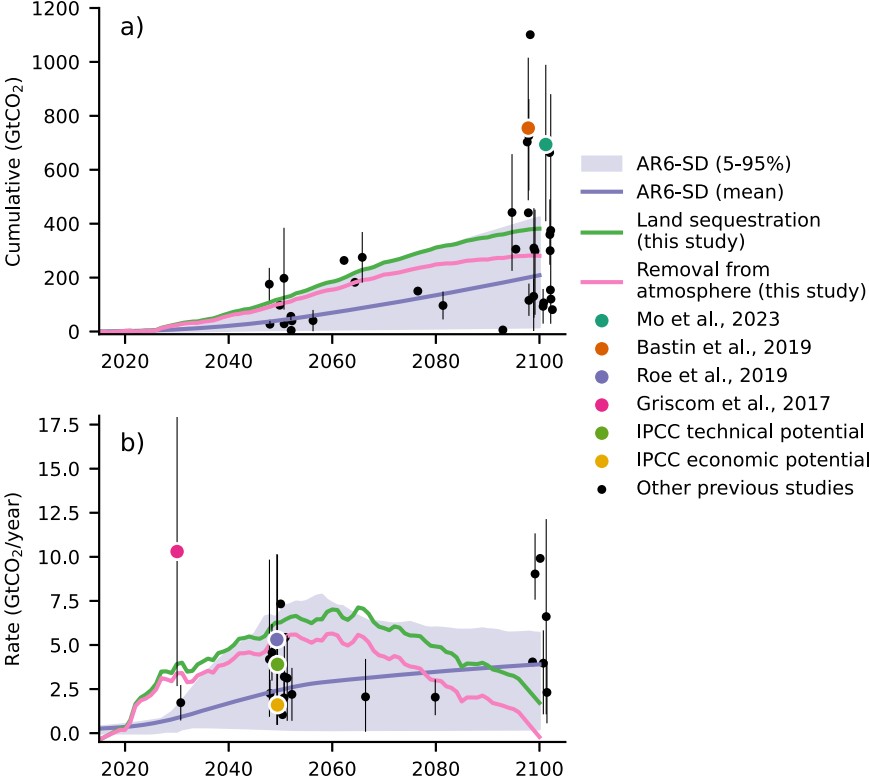

**Fig. 5 | Sequestration potentials of Afforestation/Reforestation.** Sequestration potentials of AR based on this study, previous literature, and AR6-SD are shown, expressed as (**a**) cumulative carbon sequestration through AR over time ($GtCO_2$) and (**b**) annual sequestration rate through AR ($GtCO_2$/year). The green line represents the ensemble mean (10 members) additional sequestration by land compared to REF achieved in this study and the dark red line the corresponding removal (positive) from the atmosphere, both smoothed with a Savitzky-Golay[122] filter. The 5–95% percentiles (purple shading), as well as the mean (dark purple line) of AR sequestration estimates of AR6-SD are also presented, estimated based on a subset of 300 scenarios for which explicit information on C sequestration from AR is available. The black dots (and the associated minimum-maximum ranges with vertical thin black lines–if applicable) represent (mean) AR sequestration estimates based on the compiled list of 40 previous studies (see "Methods", and Supplementary Data 1). The most widely cited studies among them[19,20,67] and the most recent estimate of Mo et al.[75] which are explicitly mentioned in the "Discussion" section, are shown with different color. The technical and economic mitigation potentials suggested by IPCC WGIII Ch.7[174] are also shown for comparison. To aid visualization, the studies have been randomly shifted in time by ±2.5 years.

## Changes in regional hydroclimate

Gridcell-level results (Fig. 6, Supplementary Fig. 6) show patches of cooling emerging as early as 2030, when the mean temperature pattern under AR is already statistically significantly different compared to REF (see "Methods", and Supplementary Fig. 7). Regions with significant cooling during 2030–2050 emerge not only for mean, but also for more extreme temperature, mostly over the northern high-latitudes, and over Latin America, and central Africa (Fig. 6, top row). Such cooling becomes progressively more robust and widespread over time, and is already dominant during the period around the peak of global temperature (2050–2070) (Fig. 6, mid row), especially over Africa, Latin America, and the northern high-latitudes. The pattern of mitigation is consistent across the mean, and the high and low temperature extremes, but differs in magnitude over the northern-latitudes where low temperature extremes are reduced more strongly. Patches of warming at the sites of forestation in Eurasia and North America emerge, which are however statistically insignificant. By the end of the century, cooling is dominant globally for mean temperature (Fig. 6, bottom row). Cooling is stronger over Africa, Latin America south of the Amazon rainforest, northern Asia, and North America. A similar pattern is obtained for the extreme high temperature, albeit with a weaker temperature mitigation over the northern high-latitudes. On the contrary, mitigation for the extreme low percentile is stronger over the northern high-latitudes, and a small patch of statistically significant warming emerges in eastern Asia over sites of forestation.

Our results further suggest a wetter hydroclimate over forested regions emerging during the mid-century (Fig. 7). This is most clearly evident with increases in relative humidity and cloudiness over sites of forestation, which start emerging already from 2030 onwards, and are especially pronounced and widespread over Latin and North America, Africa, and east Asia. This is accompanied by statistically significant but less widespread patterns of precipitation and evapotranspiration increases evident over Latin and North America, Africa, and east Asia, where the hotspots of changes in cloudiness and relative humidity are reported. Over sites of forestation reduced albedo leads to an increase in surface net radiation, thus resulting in an increase in latent heat fluxes commensurate with the reported increase in evapotranspiration and in a reduction of sensible heat flux (Supplementary Fig. 8). An increase in albedo is reported over the Arctic already from the mid-century, which can be explained by the increase in sea-ice, which is however noisy at the global scale (Supplementary Fig. 9). Surface ocean pH reaches a 0.03 increase under AR compared to REF (Supplementary Fig. 9).

## Discussion

### Mitigation potential of ambitious Afforestation/Reforestation

Here we go beyond simply using marker IAM scenarios and rather utilize an ensemble of 1259 land use scenarios, which to our knowledge is unprecedented. This allows us to navigate through the more optimistic spectrum of the wide ensemble of opportunity that IAMs can offer[49] and design an AR scenario aligned in spirit with the ambitious

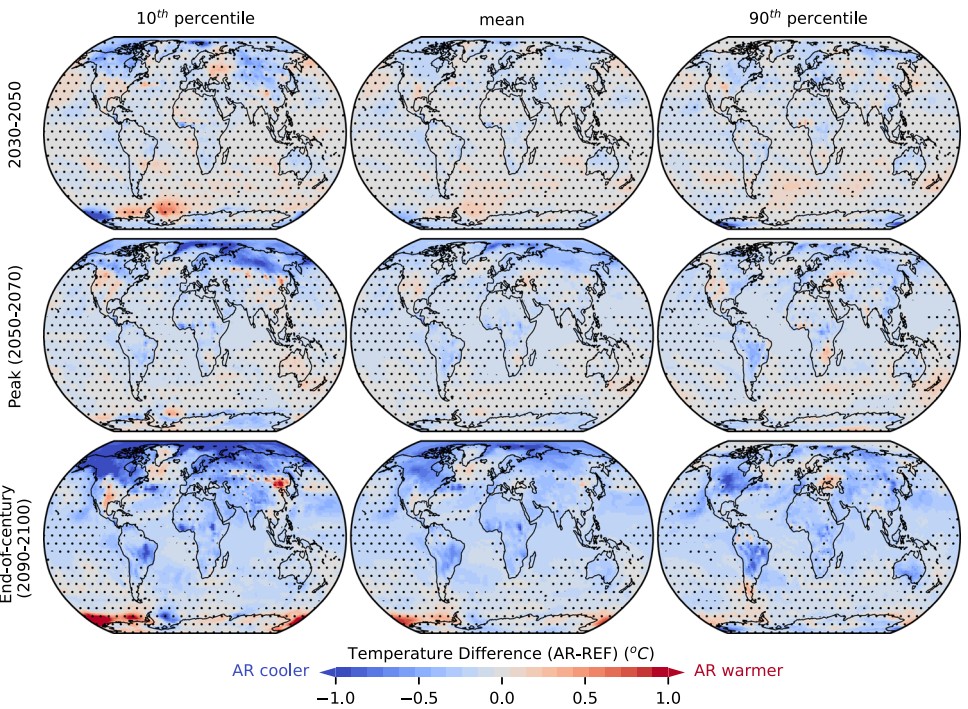

**Fig. 6 | Spatiotemporal pattern of temperature change.** The differences in 10th percentile (low temperature extremes, left), mean (mid), and 90th percentile (high temperature extremes, right) of 2 m air temperature between AR and REF simulations during 2030–2050 (top), the period around peak warming (2050–2070) (mid), and end-of-century (2090–2100) are shown. A negative difference (blue color) indicates that temperature is lower in the AR scenario. Dots indicate regions where the difference is statistically insignificant at the 5% level, estimated with a two-tailed Student's t-test after correcting for lag-1 temporal autocorrelation[123] (see "Methods").

climate policies that are necessary to overcome climate uncertainty and yield robust results in terms of achieving our climate goals[13,50]. For the first time, the mitigation potential of ambitious AR in the range of country pledges[24] is studied under an overshoot scenario with a fully coupled high-complexity ESM. This is particularly important since delayed climate action[15], insufficient Nationally Determined Contributions[12], the policy credibility gap[13], political feasibility concerns[14], and the possible overstatement of CDR potentials in IAMs[51], suggest that reaching 1.5 °C with no or limited overshoot is less likely. This increases the possibility and policy-relevance of high temperature overshoot pathways[11,12] unless immediate drastic action is taken. Our multi-ensemble member approach allows for a robust statistical treatment of the results, adding useful insight to the uncertainty surrounding the responses of the Earth system to temperature overshoot pathways[16].

Even though direct comparisons with previous studies employing different spatiotemporal AR patterns, models, and/or emission pathways cannot be made easily, our findings are in rough agreement with previous studies. Matthews et al.[40] reported a 0.02 °C reduction of peak warming and a lower efficiency of 0.03 °C for every 100 GtCO$_2$ sequestered in land, although under the SSP1-1.9 emission pathway and an idealized AR pattern. Sonntag et al.[39] reported similar results with MPI-ESM yielding a 0.27 °C end-of-century temperature reduction, as a consequence of an 800 Mha forest area increase concurrent with 100 Mha of avoided deforestation compared to the reference scenario under RCP8.5 emissions. Using a reduced complexity climate model Dooley et al.[52] demonstrated an upper limit of 0.25 °C temperature reduction in 2100, but a negligible impact on peak temperature by ecosystem restoration efforts under a low overshoot scenario. However, their scenario yields peak temperatures in 2030–2050, which is too early for the signal to emerge, since time is needed not only for the CDR to be scaled up, but also for carbon to accumulate. This agrees with our results where the signal emerges only

in 2052, despite ambitious deployment early on, and only after 132 GtCO$_2$ have been accumulated in land. Under a high-emission scenario Arora and Montenegro[53] reported a global temperature reduction of 0.25 °C estimated with an ESM, after increasing forest cover following an idealized pattern by 1010 Mha globally due to an increase of the land C sink by 440 GtCO$_2$, roughly agreeing in magnitude with our estimate. Recently, based on the LUMIP simulation employing the land use pattern SSP1-2.6 under emissions following SSP5-8.5, Loughran et al.[43] reported insignificant changes in global temperature across 6 ESMs following a land C sink increase with an intermodel range of 37–220 GtCO$_2$. However, part of this apparent uncertainty may be related to the lack of ensemble members, and the significant differences in forest area increase across models.

The extent to which our findings are model-specific remains to be assessed, by utilizing a multitude of available ESMs, each with different process representation and climate sensitivity under an emission-driven configuration. Both peak and end-of-century temperature under REF are at the lower range of the CMIP6 SSP5-3.4os multi-model ranges (5 models not including MPI-ESM) of 2–4.35 °C and 1.39–3.47 °C, respectively, as recently reported by Asaadi et al.[54], suggesting that MPI-ESM is among the models with the stronger cooling behavior as a response to negative emissions. However, direct comparisons cannot be made with confidence, since in our study land use under REF is constant, while it follows the trajectory of SSP5-3.4os in Asaadi et al.[54]. It should also be noted that the ranges reported therein are not based on multiple ensemble members. Results from the Zero Emissions Commitment Model Intercomparison Porject (ZECMIP)[55] have demonstrated that MPI-ESM shows a stronger cooling following the cessation of emissions compared to other models[56]. Despite the sensitivity of MPI-ESM to negative emissions not having been directly compared against other models so far[16,54] and uncertainty remaining, it is the additional sequestration under AR and the consequent feedbacks that determine temperature mitigation compared to REF.

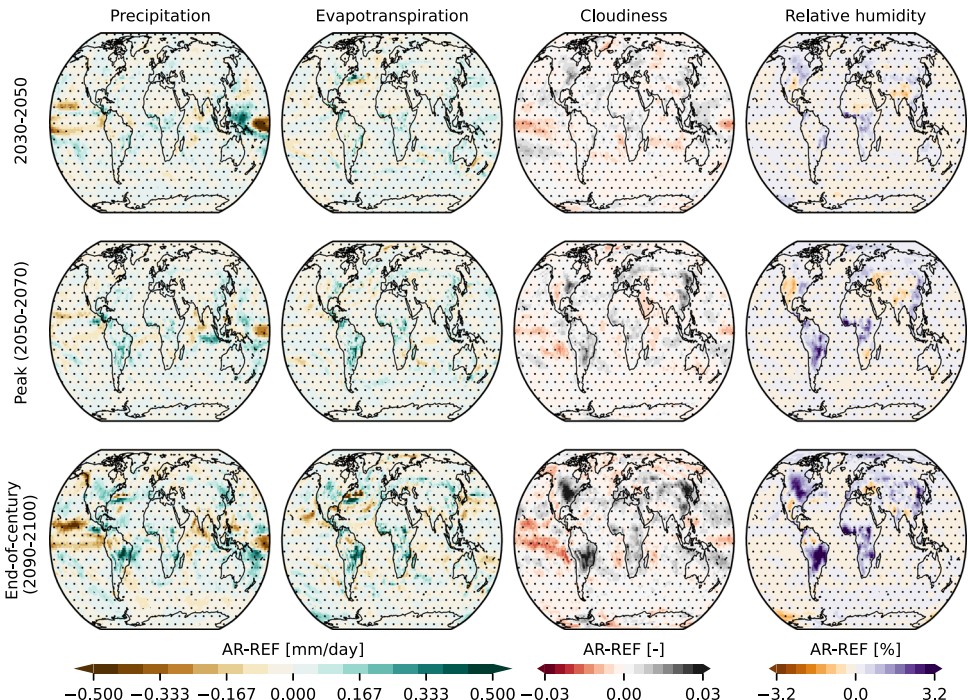

**Fig. 7 | Spatiotemporal pattern of changes in hydroclimate.** From left to right, the differences in mean precipitation (mm/day), evapotranspiration (mm/day), cloud cover fraction, and relative humidity (%) between AR and REF simulations during 2030–2050 (top), the period around peak warming (2050–2070) (mid), and end-of-century (2090–2100) (bottom) are shown. A negative value indicates a reduction in the AR scenario. Dots indicate regions where the difference is statistically insignificant at the 5% level, estimated with a two-tailed Student's t-test after correcting for lag-1 temporal autocorrelation[123] (see "Methods").

Therefore, it is rather the feedback strength within MPI-ESM that would play the key role. In particular, both land carbon gain due to increased atmospheric $CO_2$ concentration as well as land carbon loss due to elevated temperature in MPI-ESM are below the CMIP6-model average[57]. As a result, MPI-ESM is placed close to the average of CMIP6-models in terms of carbon sequestration when the total effect of elevated $CO_2$ is considered. However, MPI-ESM has also a slightly below-average Transient Climate Response compared to other ESMs[58], indicating that the cooling reported here is possibly below anticipated average ESM responses, for given levels of carbon sequestration. All this adds confidence to the robustness of our results, demonstrating the effectiveness of large-scale AR at reducing temperature even under strongly reduced emissions scenarios.

Changes in rainfall can affect terrestrial ecosystem water and carbon fluxes in various ways, depending on the ecosystem type[59]. De Hertog et al.[60] reported on the local and non-local biogeophysical effects of idealized forestation on the water cycle with the MPI-ESM and the Community Earth System Model (CESM) ESMs, with both models reporting increased evapotranspirative fluxes and precipitation over the tropics and subtropics agreeing with our results, albeit with CESM not yielding increased water fluxes over the higher latitudes. A wetter hydroclimate with increases in rainfall due to AR has also been recently reported based on a kilometer-scale model[61]. However, based on an idealized global forestation scenario with CESM Portmann et al.[30] reported that pronounced precipitation and cloud cover changes did not coincide with hotspots of forest area increase, but were rather driven by large-scale circulation adjustments.

When it comes to the obtained temperature pattern at the gridcell level, direct comparisons with previous findings cannot be made, since they significantly differ in the AR spatiotemporal pattern and other land use, and the experimental setups, whereas the number of relevant ESM-based studies is limited. For example, under a SSP5-3.4os scenario where AR is applied instead of bioenergy crops after 2040, Melnikova et al.[44] reported patches of strong warming over the mid- and high-

latitudes compared to a fixed land use scenario, even though locally at many forested gridcells cooling emerged. However, this was a concentration-driven run where the land C uptake does not result in a reduction of atmospheric $CO_2$ and thus the results reported therein are indicative of the biogeophysical effects only. De Hertog et al.[34] suggested that the total biogeophysical effect of AR in ESMs (including MPI-ESM) under idealized AR patterns tends to yield a temperature increase over the mid- and high-latitudes and a cooling over the (sub)tropics. According to our results following an emission-driven approach, this signal is dominated or at least compensated for locally by global biogeochemical cooling caused by the reduction of atmospheric $CO_2$ levels, as the land takes up more $CO_2$. This agrees qualitatively with Sonntag et al.[39] who also used MPI-ESM and is noteworthy because here we employ a scenario with stronger emission reductions and hence significantly lower $CO_2$ fertilization rates. However, the extent to which this can be consistent across models remains to be assessed.

An important emerging feature in our simulations is the early rise and continuous buildup of the land C stock, with high productivity reached already within decades, followed by a continuous forest C uptake, as the woody and soil C pools saturate only slowly[62]. However, despite the continuous increase of forest cover, the signal of AR sequestration is dominated by the emission trajectory that reaches net-negative thus drastically reducing atmospheric $CO_2$ concentrations, and consequently the strength of $CO_2$ fertilization. Nevertheless, AR offers a notable buffer delaying the transition of the land C sink from a sink to a source by ~10 years. This demonstrates that a rapid deployment of AR early on can provide a continuous C sink acting on both short and long timescales. However, atmospheric C removal is less than total land C uptake due to the emerging carbon-cycle feedbacks[63]. In particular, oceanic C uptake is dependent on atmospheric $CO_2$ partial pressure and hence part of the increase in the land C sink is compensated for by less uptake by the ocean as atmospheric $CO_2$ concentrations decrease, or even transits to being a source[64–66], as has also been reported here.

Across the literature a wide range of potentials has been reported (Fig. 5, see "Methods", and Supplementary Data 1), with our estimates of cumulative C sequestration under AR compared to REF being towards the upper range of AR6-SD (300 scenarios for which information is available) throughout the 21st century. This potential is within the upper range of estimates found in previous literature during early- and mid-century but falls within the mid-range by 2100. When it comes to yearly sequestration rates, our results lie within the upper range of AR6-SD and previous literature estimates during the early- and mid-century, before levelling off and reaching the lower-range of estimates by 2100. Notably, our estimate reaches ~50% of the estimate of Bastin et al.[67], who consider an AR area of similar magnitude (~900 Mha) yielding 752 GtCO$_2$, and received wide attention and criticism[68–74]. Similarly, our estimate reaches ~55% of the most recent comprehensive estimate of Mo et al.[75] who employed several ground-sourced and satellite-derived approaches and suggested that 694 GtCO$_2$ could be sequestered through reforestation, in agreement with Bastin et al.[67]. However, the AR pattern employed here also includes afforestation over land that is inevitably less productive, which could partially explain this difference, while it should be noted that the estimates of Mo et al.[75] are based on current climatic conditions. Our estimate remains within the lower boundary of the 2.7–17.9 GtCO$_2$/year range reached by 2030 reported by Griscom et al.[19], who however consider a potential reforestation area up to 1779 Mha in their calculations, which could partly explain the divergence. Roe et al.[20] have reported a potential from 0.5 GtCO$_2$/year, which corresponds to a low ESM-based estimate, up to 10.1 GtCO$_2$/year, which includes global reforestation constrained by food security and biodiversity based on Griscom et al.[19], and on average aligns with the results reported here.

It should be noted that literature estimates are the outcome of different combinations of available land, temporal dynamics of AR deployment, emission trajectories, and assumptions with regards to the capacity of forests and soils to sequester C across different biomes, and hence such a wide range can be expected[7,19,20,67,76,77]. Most importantly, only 2 of the studies compiled in Fig. 5 employ a fully coupled ESM in an emission-driven setup that accounts for climate and carbon cycle feedbacks[39,53], whereas the vast majority includes either offline estimates of sequestration potentials and thus is not easily relatable to changes in atmospheric CO$_2$ concentration and climate[19,67], or IAM-based estimates (the AR6-SD scenarios, and e.g., ref. [76,78]). IAMs have different configurations and represent the climate and carbon-cycle feedbacks with different levels of complexity[79], but they lack the detail of process representation available in ESMs, which is necessary to robustly capture the efficiency of AR, especially when it comes to carbon fluxes under overshoot scenarios[80].

Even though pooling multiple AR6-SD scenarios together eliminates the bias of selecting a single IAM scenario, and utilizing regional data allows for the regional-level dynamics to be preserved, it should still be acknowledged that with the approach employed here, the internal consistency found in single IAM scenarios is inevitably lacking. Disaggregating to the gridcell level based on biodiversity and restoration potential maps, not implementing land use transitions other than AR, and independetly selecting an emission trajectory as is done here can also break internal IAM logic. For example, while opting for the SSP5-3.4os emission trajectory allows for a robust assessment of temperature overshoot dynamics, achieving net-negative emissions in SSP5-3.4os is heavily based on carbon capture and storage (CCS)[16]. This suggests that additional land pressure would be exerted by the need for large-scale bioenergy CCS application, thus increasing competition with AR over land. However, it should be noted that inconsistencies are introduced even when a single IAM scenario is run with an ESM, due to differences in spatial resolution, land use patterns, and discrepancies in the representation of the biosphere and the carbon cycle[81,82].

## Contextualizing ambitious Afforestation/Reforestation

The present study does not propose a spatiotemporal AR pattern to be followed, but rather aims at overcoming the limitations of highly idealized patterns and of scenarios employing only moderate levels of AR that fall short of the range of total country pledges, by harnessing a multitude of available IAM-generated scenarios coupled with restoration potential maps and biodiversity proxies[34,38–41]. Using the AR6-SD ensemble allows us to consider constraints of land competition, and technoeconomic, environmental, and societal feasibility, to the extent that those are accounted for in the IAMs, and the scenario here is hence spatiotemporally plausible only to the extent that IAM scenarios can be treated as such. Concerns regarding the uncertainties and weaknesses of IAMs have been reported[83], such as their lack of social and institutional considerations, and their inability to conceive more radical societal reorganization and policy challenging strategies, thus not being able to capture the full socioeconomic possibility space[84–86]. However, it should be noted that the IAMs still remain the main tool at hand when it comes to contextualizing the technical, social, and economic developments in the world, which is necessary for developing a constrained AR pattern that goes beyond fully idealized setups.

At the same time, it should be acknowledged that future AR will inevitably include reversing historical deforestation, not only because such regions can naturally support growth, but also if afforestation is to be avoided[87]. This increases the relevance of patterns aimed at prioritizing reforestation, as is the case here. Nevertheless, the feasibility of setting such an ambitious global AR target in the range of country pledges can be questioned[24,25,52,88]. Overall, the pledges by 9 countries amount to 90% of total AR pledges globally (Supplementary Fig. 1), including the United States (25%), Canada (4%) and China (4%), whose net-zero targets are considered of lower confidence rating, and Saudi Arabia (42%), India (4%), Ethiopia (4%) and Australia (4%), whose net-zero targets are of much lower confidence rating based on Rogelj et al.[13]. The scenario employed here shows less concentration of AR with 39 countries amounting to 90% of the global target (Supplementary Fig. 1). However, this also relies on countries whose net-zero targets are considered of lower (United States, China, Russian Federation, Colombia) and much lower (Brazil, Argentina) confidence rating[13]. This credibility issue can be even more alarming when one considers that total pledges would likely grow in the future, as more countries set their long-term strategies[25].

In our scenario grazing land is significantly reduced and no deforestation occurs, possibly implying a strong dietary shift and reduction of meat consumption[89], coupled with a sustainable intensification of remaining grazing lands and regulations to prevent deforestation elsewhere as a compensating mechanism for lost agricultural land[90]. This fits with a land-sparing paradigm that can indeed offer high mitigation potentials[91] and benefit biodiversity[92,93]; however, the extent to which this can be considered realistic from a socioeconomic perspective is not investigated here. It should also be noted that given fair burden sharing considerations, our scenario, which largely includes forest expansion over the global South, would require financing AR deployment over these regions by the more developed countries[94], as is also suggested by the pledge of Saudi Arabia. Until this day, lack of public and private finance has been one of the key barriers to meeting global restoration needs, and scaling-up funding would be a big challenge[95].

Socioeconomic factors could also possibly pose barriers to AR implementation or a threat to the permanence of a newly planted forest, while the possible societal consequences and associated tradeoffs should also be considered[45,96]. In particular, the rates of sequestration achieved here can carry a high risk of exceeding sustainability thresholds[97], while the strong dependence of IAMs (and consequently of our scenario) on land-based mitigation in the Global South to reach ambitious climate targets might pose high risks to food security and raise equity concerns[88]. Comparing the AR pattern

employed here with socioeconomic factors such as poverty, population density, governance, land tenure, and indigenous and community land is thus crucial (see "Methods", and Supplementary Fig. 10). In particular, 6.5% (61 Mha) of the AR in our scenario is applied over gridcells with a significant extent of indigenous and community land (>30%). Planting forests in these regions can often violate the will of indigenous people who can be strongly tied to their land spiritually, financially, and/or are nutritionally dependent on local food production[24], and lead to forced physical or economic displacement[98]. At the same time, 17.6% (165 Mha) of AR is deployed over regions where more than 25% of the population live below the international poverty threshold. Implementing AR over poverty-stricken regions can carry the risk of depriving people of their livelihoods and exacerbating poverty, even though positive outcomes on livelihoods have also been reported[99–101]. Additionally, 2% (19 Mha) of AR is applied over regions with a population density higher than 500 people/km$^2$ where rates of human disturbance and deforestation can be higher[102,103], thus likely jeopardizing permanence.

At the same time, 18% (169 Mha) of AR is applied over regions with poorer governance (governance indicator <0.3), and 16% (151 Mha) over regions with insecure land (>50% of survey respondents perceiving their land or property as being insecure). Poor governance and weak rule of law are considered to be significant barriers to successful implementation and permanence of AR, facilitating forest loss and degradation[78,104]. Land tenure insecurity can significantly threaten the establishment and permanence of a newly planted forest[105], since clear and secure land tenure can determine farmers' decision to adopt reforestation and maintain a forest over the long-term[101,106].

## Outlook

Societies have been (re)planting trees for centuries, and AR has reasonably emerged as an efficient way to naturally store C[6], while at the same time restoring ecosystems that offer a wide variety of services to our communities and support life[107]. Our results based on MPI-ESM clearly demonstrate that ambitious AR in the range of country pledges can mitigate a temperature overshoot by lowering global peak temperature, overshoot duration, and end-of-century temperature. Temperature is mostly reduced locally at the sites of forestation, suggesting that the overall biogeochemical cooling effect of the increased land C sink can dominate, or at least compensate for any biogeophysically-induced warming that can arise. Mitigation emerges both for average and for extreme temperature, yet to infer upon the appropriateness of large-scale AR, one should also consider potential AR-induced changes in climate hazard exposure, such as droughts[108], rainfall variability and extremes[59,109], heat stress[110], and compound events[111], that could increase adaptation needs locally or regionally[112]. However, such a multi-dimensional investigation is beyond the scope of the current study and could be the focus of future work, potentially also employing state-of-the-art high-resolution convection-permitting models that can take into account the effects of fine-scale topography, and the local features of atmospheric convection on climatic extremes[61,113]. Future work could also focus on investigating to what extent the model behavior reported herein can be model- or pattern-specific by comparing multiple ESMs and AR patterns under an emission-driven configuration.

The mitigation potential demonstrated here constitutes ambitious AR as a useful complementary short- and long-term mitigation tool for climate action even under a scenario with strongly reduced emissions, where fertilization of vegetation by CO$_2$ is weaker. However, the scale of mitigation achievable by such high-ambition AR scenarios—with 0.08 °C and 0.2 °C decrease of global mean peak and end-of-century temperature respectively—clearly shows that AR does not alleviate the need for high ambitions in emission reduction[52]. Importantly, even though a normative judgment on the desirability of ambitious AR is not made here, the results demonstrate the possible socioeconomic tradeoffs

associated with it, as well as the significant barriers to implementation and possible threats to permanence that exist.

## Methods

### Model and experimental setup

We run fully coupled ESM simulations using the CMIP6 configuration of MPI-ESM (MPI-ESM-1-2.01p7-LR)[114]. MPI-ESM and its land surface component JSBACH have been widely applied, evaluated, and compared against observations and other models, and have been generally found to perform well for various key land surface variables[115–117], and for both biogeophysical and biogeochemical effects[35,118]. The model is run in an emission-driven setup, which means that atmospheric CO$_2$ concentration is not prescribed, but is rather dynamically updated. The fully coupled emission-driven setup offers a dynamic interaction of water, carbon, and energy fluxes between the land, the atmosphere, and the ocean, thus allowing for a full representation of the complex biogeophysical/biogeochemical effects and feedbacks of AR within the entire Earth system.

Our experimental setup includes an AR and a reference (REF) scenario spanning from 2015 to 2100, following SSP5-3.4os, except for land use. SSP5-3.4os reaches a total radiative forcing of 3.4 W/m$^2$, and is an overshoot scenario that follows SSP5-8.5 emissions until 2040, followed by rapid decarbonization, thus reaching net-zero around 2070 and then net-negative (-−3.8 GtCO$_2$/year)[16,119]. To fully isolate the total effects of AR on the Earth system, which is estimated as the mere difference between the AR and REF scenarios, we keep land use constant at the 2015 state under REF. Hence, we do not include any avoided deforestation, or other land use transitions, which would further complicate the accurate estimation of CDR. Towards that direction, even though increased forest area and wood biomass availability could impact the demand for wood products and hence harvest rates via market mechanisms[120], we keep wood harvest amount constant at 2015 levels for both scenarios. At the same time, the amount of vegetated surface is kept constant for both scenarios, and competition among different Plant Functional Types (PFTs) over vegetated land is switched off, thus not allowing forest to expand as a response to a warming climate, in contrast to the CMIP6 model configuration. This allows for prescribing forest area in each timestep, thus being able to fully isolate the signal of AR, which is the main aim of this study. First, accounting naturally expanding vegetated land could raise a concern as to whether CDR can be claimed over land that becomes available due to our sole failure to decarbonize our economy. Second, the permanence of stored carbon over such land is in fact questionable since, assuming further climate mitigation, forest would contract again, suggesting carbon fluxes towards the atmosphere. We run in total 20 model runs; 10 ensemble members for each scenario, to allow for a robust probabilistic treatment of our results.

### Probabilistic treatment

To robustly assess the global dynamics of the overshoot and its characteristics, a probabilistic framework is employed. To estimate the time of signal emergence, the overlapping coefficient (OVL)[121] is employed, which is a measure of similarity between the probability density functions (pdfs) of two populations f1($x$) and f2($x$), and is calculated as follows:

$$OVL = \int \min[f1(x), f2(x)]dx \qquad (1)$$

Over a temporally moving window, global mean 2 m air temperature data from all ensemble members are pooled together, creating two samples, whose OVL is estimated with a gaussian kernel density function. This process is repeated by bootstrapping 1000 times, and a mean OVL is estimated. To estimate an OVL threshold below which dissimilarity between AR and REF distributions can be

assumed, an autoregressive model (AR (1)) fitted to REF global temperature is used. At every temporal window, first 1000 pairs of stochastically generated temperature timeseries of trend and sample size equal to REF are created, for each of whom OVL is then estimated. The lower 5th percentile of the 1000 OVL values estimated from the synthetic data (coming from the same known AR(1) process) is determined as the OVL threshold, below which two data samples can be considered to come from dissimilar distributions (at the 5% significance level). The year of signal emergence is then calculated as the year when the mean OVL between REF and AR data starts becoming consistently lower than the significance OVL threshold. This process is repeated with a window length ranging from 5 to 10 years and a mean year of emergence is estimated, to make sure that the choice of window length does not bias the results.

To robustly characterize end-of-century temperature while taking into account interannual variability, the full end-of-century 5-year period (2096–2100) is considered. Similarly, to get robust estimates on peak temperature and avoid biases by likely extremely hot or cold years later or early on, the full 5-year consecutive period yielding the highest average temperature for each ensemble member is considered, and the temperature data are then pooled together. The choice of the period length does not bias the results, as shown in Supplementary Fig. 11.

To robustly capture overshoot duration, the signal needs to be isolated from the noise of internal variability. Hence, a smoothing Savitzky-Golay[122] filter is applied. Overshoot duration is then estimated with a varying smoothing window length, ranging from 5 to 10 years, to alleviate any bias induced by the choice of window length. This process is repeated 1000 times, where each time a random combination of 10 ensemble members with replacement are chosen. The mean overshoot duration is then estimated as the mean of the values obtained by bootstrapping. To get a clearer picture of the responses of the full spectrum of temperature variability, different percentiles of global average daily temperature are treated with the same approach.

At the grid level, the statistical significance of changes in 2 m temperature (Fig. 6) or other hydroclimatic variables (Fig. 7, Supplementary Fig. 8) during different time periods is inferred at the 5% level. To do this, for each scenario yearly values from all 10 ensemble members during that period are pooled together, and a two-tailed Student's t-test adjusted to account for lag-1 temporal autocorrelation[123] is applied. To assess field significance (i.e., whether the annual average temperature field under AR is statistically significantly different from REF) the probability of at least one false positive among the multitude of gridcell-level tests is accounted for[124]. If a single gridcell is found to be statistically significantly different after accounting for this probability, then the global null hypothesis that a statistically significant difference between the AR and REF temperature field does not exist can be rejected[125]. To increase the robustness of the approach and test its sensitivity, a) temperature data at the gridcell level are pooled over a moving window with length ranging from 1 to 10 years, and b) field significance is declared when the null hypothesis is consistently rejected for a consecutive number of years ranging from 1 to 5, defining as year of emergence the starting year of that period. The results are presented in Supplementary Fig. 7.

## AR scenario development

To design an ambitious constrained AR scenario following technoeconomic, environmental, and societal constraints, we leverage (a) the AR6 Scenarios Database (AR6-SD)[47], (b) the restoration potential map published by Griscom et al.[19] (hereafter GRS), (c) the Atlas of Forest and Landscape Restoration Opportunities (hereafter ATL)[126], and (d) the Very Low and Low Human Influence maps published by Riggio et al.[127] (hereafter VLHI and LHI, respectively). All maps are regridded to the MPI-ESM resolution with a conservative remapping algorithm.

The AR6-SD includes a multitude of IAM-generated scenarios that consider the pressure exerted on land by rising population and food demand, carbon pricing strategies, energy policies and economic costs[47]. Here, we use the 1259 scenarios for which explicit information on global forest area is available to capture the feasible range of global AR[79]. The change in forest area, when positive, can represent not only active AR deployment, but also abandonment of agricultural land, and natural regrowth and succession dynamics, depending on the IAM[20]. For each year, we estimate the forest area change compared to the previous year for each scenario, and pool all the values together. To obtain an ambitious AR target at the global level which is in the range of estimates of total country pledges[24,25] we use the 90th percentile of the pooled global yearly forest area change estimates, which cumulatively reaches 595 Mha by 2060, and 935 Mha by 2100 (Fig. 1). This forest area increase would roughly correspond to its complementary threshold, the 10th percentile of cropland and pasture area change, which suggests that AR is mostly applied at the expense of grazing lands across AR6-SD, while croplands remain relatively stable (Fig. 1). To mimic that behavior, AR only replaces grazing land in our scenario. It is important to note that we do not select individual scenarios, but rather a high percentile across all values each year, and that we refrain from making any judgment upon the likelihood of the individual AR6-SD scenarios, which are treated as different equiprobable narratives across the feasibility space[79], nor do we consider the high percentile utilized here as being less probable than the mean[49].

Given the lack of pledges for many countries globally, and that the significantly high pledge of Saudi Arabia (42% of global) can only be met through international offsets in addition to domestic land-based CDR[25], the spatial disaggregation of global AR pledges is not made based on the individual country pledges themselves, but rather regionally based on AR6-SD estimates. Unavoidably, this creates a spatial mismatch between the scenario employed here and pledges at the country level (Supplementary Fig. 1). However, it should be noted that employing the AR6-SD allows for an explicit treatment of the temporal evolution of AR across the century based on technoeconomic considerations, instead of simply interpolating pledges up to 2060, and arbitrarily extrapolating thereafter.

Here, we use information available in AR6-SD to disaggregate the global yearly AR target across the 5 world economic regions shown in Fig. 3; a regionalization typically used in IAMs. Even though different regionalizations are also available in AR6-SD, the one chosen here offered the most available scenarios (1124). In a similar fashion, for every region we pool available scenarios together and select the 90th percentile of the pooled regional yearly forest area change. Given that a different number of scenarios is available at the regional compared to the global level, and that a percentile for every region is chosen rather than a specific scenario consistent across regions, we rescale the regional estimates so that their sum matches the yearly global target (preserving their relative magnitude with respect to their sum). Interpolation of the 5-year AR6-SD data to yearly both at the global and regional level is performed with a piecewise cubic hermite interpolating polynomial algorithm.

The regional-scale yearly AR target is further distributed spatially, from regional level to the MPI-ESM grid by following the GRS and ATL maps. We consider the ATL restoration potential that includes reforestation and tree integration into mixed-use landscapes and excludes croplands[126]. GRS is a subset of ATL further constrained by food security and biodiversity concerns and excludes a range of areas from AR: croplands, regions where forest is not the native cover type (afforestation), regions with dense rural population, regions with intensive management, and boreal regions that could potentially yield strong local warming[19]. Both GRS and ATL potentials per gridcell are expressed as fractions of the gridcell area. Using these maps to guide our scenarios takes implicitly into account the considerations therein. For a given year and region, the AR target is first distributed across

gridcells prioritized by GRS. The gridcells considered for AR are the ones where (a) grazing land is available, (b) the remaining GRS potential is greater than zero, (c) forest area has not already increased more than 10% in that year, to avoid excessive AR rates that could be institutionally challenging or incompatible with sustainable local socioeconomic changes, and (d) at least 20% forest cover has historically been or could potentially be sustained, to not use climatically unsuitable regions that would largely require management practices. The target is uniformly applied to the gridcells considered; however, the amount of AR applied over each is limited by (a–d). AR applied within each step reduces the remaining GRS potential by the same amount. This is an iterative process until either the regional target is met, or there are no remaining gridcells to be considered because (a–d) are no longer satisfied anywhere in the given region. In the latter case, the remaining AR target is filled by gridcells prioritized by ATL with a similar iterative process, and if still needed, the rest of the gridcells (satisfying (a–d)) can also be used to meet the AR yearly target. As a result, the AR pattern is constrained first by GRS (362 Mha in the resulting pattern), and then by ATL (375 Mha), but meeting the regional AR target each year can also mean increasing forest cover beyond total restoration potential (GRS and ATL combined) and reaching other gridcells (not prioritized by GRS or ATL), if needed (198 Mha). Reforestation of historically deforested sites is thus strongly prioritized in our approach, but reaching an ambitious target while preserving croplands globally constitutes afforestation (i.e., increasing tree cover beyond 1850 levels) and utilization of land not prioritized by restoration potential maps inevitable.

To account for biodiversity protection, heavily managed pastures are replaced first within every iterative step of the distributive process, and then the more lightly managed rangelands are replaced progressively, starting from the less to the more biodiverse. To protect old-growth grassy biomes which can be of huge ecological importance and exhibit high biodiversity richness[87], we characterize rangeland biodiversity by using the VLHI map to determine which rangelands can be considered as being closer to a pristine, and thus as more biodiverse state, and exclude them. Natural grassy biomes which are not characterized as grazing land are also excluded. Then, we use the LHI map to split remaining rangelands into 4 biodiversity groups, based on the level of agreement among the datasets used by Riggio et al.[127]. By using these maps as a proxy for biodiversity, we assume that the higher the dataset agreement is, the more likely a gridcell experiences low human influence, and hence it is more likely less heavily managed, with less infrastructure, human stress, and cattle densities, weaker land fragmentation, and thus, probably closer to its pristine state, and hence likely more biodiverse[127]. Even though anthropogenic disturbance is known to be a major driver of biodiversity loss[128], biodiversity richness is not necessarily associated with the level of human influence[129]. At the same time, although abandoning pastures can promote biodiversity, in some cases this does not translate to ecosystems returning to their initial biodiverse state[130]. Even though we acknowledge these shortcomings, a more detailed treatment of biodiversity concerns is beyond the scope of this study, and the usage of largely available metrics of human influence is preferred here[127].

Evidently, in our study no biophysical or climatic limit is applied when considering gridcells eligible for AR, other than whether forest cover has or could potentially (based on MPI-ESM estimates) be at least 20%. Stricter limitations would reduce the amount of grazing land used for AR, since grazing land is often found in semi-arid, transitional regimes (Fig. 2). Hence, increasing forest cover over such areas possibly implies the need of some form of forest management in the real world to support forest growth. Still, carbon sequestration is climate-dependent in MPI-ESM, and more arid regions typically yield lower productivity and sequestration rates (Fig. 4c), and hence afforesting more arid rangelands would not contribute to the mitigation potential in the same way as the more productive areas included in our scenario.

It should also be noted that our simulations start in 2015 and do not include historical land use transitions having occurred later than 2014. According to FAO data[131], forest area has decreased by ~30 Mha from 2015 to 2021. By that time in our model AR amounts to ~5 Mha, which corresponds to ~0.5% of total AR application, thus yielding a net difference of ~35 Mha of forest area. By 2021 land C stocks are increased by ~3 GtCO₂ (0.81 PgC) in the AR compared to the REF scenario, constituting 0.78% of total additional sequestration (382 GtCO₂). Clearly, the bulk of forest area increase and additional C sequestration under AR occur after 2025 in our simulations, which suggests that our results remain relevant with respect to overshoot dynamics and mitigation, regardless of this discrepancy with recent historical data.

An overview of the methodology used to develop our scenario is shown in Fig. 2 and the detailed resulting pattern is presented in Fig. 3 and Supplementary Figs. 1 & 2.

To test the sensitivity of the AR scenario development to the different constraints employed, different configurations are considered. In particular, we develop scenarios where: a) only the restoration map published by Griscom et al.[19] (GRS) and the Atlas of Forest and Landscape Restoration Opportunities[126] (ATL) map (i.e., no explicit biodiversity consideration), b) only GRS, and biodiversity maps[127], and c) no restoration or biodiversity maps are used to constrain the scenario development.

Results are shown in Supplementary Fig. 12. Even though differences arise between the configurations, the patterns share some similarity with the AR scenario employed in this study. It should be noted that pastures are an integral part of both GRS and ATL (the first being a subset of the latter). As a result, prioritizing pastures can partly compensate for not using restoration potential maps to guide AR, due to the inherent consistency between the two. At the same time, consideration of biodiversity does not affect the partitioning between rangelands and pastures that are given up, since biodiversity is only considered for rangelands. Therefore, since pastures are generally given up first in our algorithm regardless of the configuration tested, the rough AR pattern is not heavily sensitive to the different constraints employed. However, considering biodiversity affects the spatial pattern of AR, mostly since it determines the regions where rangelands can be considered closer to a pristine state, and thus excluded from AR. Even though the AR pattern by 2100 unavoidably converges to the available grazing land, the differences across time can be more pronounced as the partitioning between pastures and rangelands, and the specifics of each configuration change.

## Literature estimates of sequestration potential

In Fig. 5, the AR sequestration estimates demonstrated here are compared with potentials that have been reported in previous literature. The list of 40 studies presented includes (a) a subset of the comprehensive list of studies provided by "CO₂ removal.org"[5,7,132] for which explicit quantitative information on AR sequestration potentials at the global scale is provided (19 studies in total)[19,39,76,133–148], (b) studies more recent than[5,7,132], which are not included in the comprehensive list (16 studies)[20,52,67,75,77,78,149–158], and (c) studies published prior to[5,7,132] that we identified as missing from the comprehensive list (5 studies)[53,159–162]. The complete list of studies including the estimates of AR sequestration potentials are listed in Supplementary Data 1.

Across the identified studies we have included only the ones for which cumulative sequestration AR potentials over a defined period, and/or instant sequestration fluxes at a defined year or period can be confidently estimated, based on the information available in each study. Studies for which sequestration potentials cannot be attributed to an increase in forest cover and mostly include other nature-based climate solutions or contain only regional estimates have been omitted. Importantly, modeling studies that are employing idealized AR patterns[34,163–165] are not considered.

For studies reporting multiple sequestration potentials, we define the range of estimates as [minimum-maximum] and calculate the mean potential across all values. For studies providing a range of values, we also estimate their mean. It should be noted, however, that not all estimates are entirely independent from one another, but quite a few studies rather build upon prior knowledge provided in the literature. As a result, the pool of studies is not treated as a statistical sample of independent values, and hence descriptive statistics (e.g., mean, standard deviation) are not employed.

The AR6 Scenarios Database (AR6-SD) estimates of AR sequestration potentials is based on a subset of 300 scenarios for which: a) explicit information on AR sequestration is available, b) the three exclusion criteria defined by Prütz et al.[166] are met. Therefore, this pool of available scenarios includes only a subset of the 1259 scenarios used to guide the spatiotemporal design of the AR pattern of this study.

### Socioeconomic indicators

To aid in the discussion of the possible socioeconomic risks and trade-offs associated with the implementation of the AR scenario employed in this study, several spatially explicit socioeconomic indicators are used. In particular, we use:

a. Governance indicator: The Worldwide Governance Indicators (WGI) provided by the World Bank are used[167]. The WGI includes country-level indicators on voice and accountability, political stability and absence of violence/terrorism, government effectiveness, regulatory quality, rule of law, and control of corruption over the period 1996–2022. Missing values are filled by taking the arithmetic mean of the specific world subregion that a country belongs to, based on the 22 geographical subregions defined in the United Nations Geoscheme by the United Nations Statistics Division. Following Andrijevic et al.[168] a composite governance indicator is estimated by taking the arithmetic mean of the six WGI components for 2022. The composite indicator is found to correlate well (>0.8 correlation) with all of the underlying components.

b. Land tenure insecurity indicator: The Prindex 2020 global dataset[169,170] of land tenure insecurity is used. This indicator describes land tenure and property insecurity as perceived by individuals over 140 countries, expressed as a percentage (%) of the survey's respondents. Missing values are filled as described above.

c. Poverty indicator: The poverty headcount ratio (%) at $2.15 (2017 purchasing power parity, lineup est. of 2019) from the Global Subnational Atlas of Poverty (GSAP)[171] is used. The data are regridded to the resolution of MPI-ESM with conservative remapping. Missing values are filled as described above.

d. Indigenous and community land: Indigenous and community land data from Landmark[172] are used. The data are regridded to the resolution of MPI-ESM with conservative remapping. In particular, the land that is considered here consists of:
   i. Indigenous land—Acknowledged by government: Documented and not documented.
   ii. Indigenous land— Not acknowledged by government: Held or used with formal land claim submitted and held or used under customary tenure.
   iii. Community land—Acknowledged by government: Documented and not documented.
   iv. Community land—Not acknowledged by government: Held or used with formal land claim submitted and held or used under customary tenure.

e. Population density: The WorldPop population count[173] is used. The data are regridded from the ~1 km resolution to the resolution of MPI-ESM by summing, and are then divided by gridcell area to obtain population density.

An overview of the socioeconomic indicators overlaid with the AR pattern, and some indicative estimates of how AR is distributed across regions with various levels of governance, land tenure insecurity, poverty, perscentage of indigenous and community land, and population density are presented in Supplementary Fig. 10.

### Reporting summary

Further information on research design is available in the Nature Portfolio Reporting Summary linked to this article.

## Data availability

The AR6 Scenarios Database (AR6-SD)[47] is freely available at AR6 Scenario Explorer and Database hosted by IIASA. The Atlas of Forest and Landscape Restoration Opportunities[126] is freely available at https://www.wri.org/data/atlas-forest-and-landscape-restoration-opportunities. The (Very) Low Human Influence maps of Riggio et al.[127], and the restoration potential map of Griscom et al.[19] are freely available with the respective publications. The Worldwide Governance Indicators (WGI) are freely available at https://www.worldbank.org/en/publication/worldwide-governance-indicators[167]. The Prindex 2020 global dataset is freely available at https://www.prindex.net/data/[170]. The poverty headcount ratio is freely available at https://pipmaps.worldbank.org/en/data/datatopics/poverty-portal/home[171]. The Indigenous and community land data are freely available at https://www.landmarkmap.org/[172]. The WorldPop population count is freely available at https://www.worldpop.org/[173]. A repository with data supporting this publication is published in Zenodo at: https://doi.org/10.5281/zenodo.12533125.

## Code availability

The Max Planck Insitute's Earth System Model (MPI-ESM-1-2.01p7-LR) is made available under a version of the MPI-M software license agreement (the license and information on how to access the code can be found here: https://code.mpimet.mpg.de/projects/mpi-esm-license). Python 3.11.2 has been used for all data analysis.

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

## Acknowledgements
Yiannis Moustakis would like to acknowledge the contributions of Raphael Ganzenmüller and Wolfgang Obermeier, whose comments have helped improve this work. This work was funded by the German Federal Ministry of Education and Research (BMBF), projects CDRSynTra (No. 01LS2101A) and STEPSEC (01LS2102A). This work used resources of the Deutsches Klimarechenzentrum (DKRZ) granted by its Scientific Steering Committee (WLA) under project ID bm1241.

## Author contributions
**Yiannis Moustakis**: Conceptualization, Methodology, Software, Validation, Formal analysis, Investigation, Data Curation, Writing—Original Draft, Review & Editing, Visualization **Tobias Nützel**: Methodology, Software, Validation, Data Curation, Writing—Review & Editing **Hao-Wei Wey**: Methodology, Writing—Review & Editing **Wenkai Bao**: Data Curation, Writing—Review & Editing **Julia Pongratz**: Methodology, Resources, Writing—Review & Editing, Supervision, Project administration, Funding acquisition.

## Funding

## Competing interests
The authors declare no competing interests.
