## [Peer Review File · Nature Communications]

Temperature overshoot responses to ambitious forestation in an Earth System ModelREVIEWER COMMENTS

Reviewer #1 (Remarks to the Author):

The authors investigate an ambitious but plausible AR global scenario in line with country pledges. and then assess to what degree a temperature overshoot could be reduced.

They develop an AR scenario reaching 595Mha of AR in 2060 and 935 Mha by 2100, based on an anormously large scenarios dbs.

They then run fully coupled simulations with the Max Planck Institute's Earth System Model (MPI-ESM).

the work as such is impressive, but I believe it is not novel enough for a Nature publication.

Biogeochemical cycles or PNAS may be a better suited avenue.

As they also show themselves, many of such global scenarios exists and have been assessed in terms of carbon effects. many were also gross overestimates (Griscom et al, Bastin et al.), not taking into account either food production needs or biodiversity effects.

This is now not changed in present study. However they align the results also in terms of reducing overshoot within the possibilities of the Max Planck Model. but e.g. perverse effects in terms of biodiversity are not assessed. they still come with a very high required area 50--900 Mha.

As such a solid study, well written. but I think better suited for journals as mentioned above

Reviewer #2 (Remarks to the Author):

Summary and main comments:

This work discusses the effectiveness of forestation in an overshoot scenario, a storyline of future global average surface-near temperature rising above common set targets (e.g., the 1.5°C target in the Paris Agreement) and subsequently falling below, typically within the 21st century. In a single-model ensemble simulation setup, the fully coupled MPI-Earth System Model is used to compute a reference (REF, no forestation) and a forestation (AR) climate scenario. The author's main claim that such land cover and land use change (LULCC) can mitigate temperature overshoot in the model in terms of height and duration seems to be well-founded regarding presented scenario design and analysis. Another ambition of the study is to apply a "plausible" AR scenario compatible with global scale AR pledges. This is done by combining scenarios from Integrated Assessment Models (IAMs) with restoration potential and biodiversity maps. The study provides useful information on the mitigation potential of AR considering overshoot, however, I have some comments that should be addressed.

I would appreciate a more in-depth analysis or discussion of the processes at play. The study mostly offers assessment from the narrow angle of global averages and sums rather than of region-specific regimes of system behaviour. This is incomplete concerning regional effects and extreme climate behaviour, which can be particularly relevant for the biophysical effects. Looking at averages might be enough to judge AR's overall effectiveness for long-term earth system behaviour, e.g. concerning impacts on the cryosphere, information which partly might be readily available from the performed coupled model simulation. However, a discussion of this relevance also seems to be lacking. If "ambitious", i.e. global-scale forestation can mitigate modelled temperature overshoot, what does this mean for the oceanic, cryospheric, and land-bound systems? In that sense, what are the significant novel implications for climate science and/or climate action?

Another concern related to this is how time of signal emergence, peak temperature difference and interannual variability are treated. Concerning peak global mean 2m T, I understand that a robust comparison needs some smoothing of the interannually fluctuating behaviour of global mean T. However, I think it is important to not overlook differences in extremes induced or enhanced by interannual variability. E.g., how does the running window 10th and the 90th percentile compare between AR and REF? Here I think some work either with statistical or more process analysis is useful to give a less simplified picture of the Earth System behaviour. This will also help in the following discussion, what does it mean to find the two scenarios distinguishable? Can you provide examples for contexts in which 0.09°C less globally make a significant difference regionally for certain processes?

My second comment addresses "plausible" scenario construction. As the authors mention, their judgment of plausibility is based on fitting the AR trajectory to globally aggregated AR pledges and doing so by finding a fitting percentile in AR areal amount within the IAM scenario collection. Additionally, they base the distribution of these area indications across global land on a method that includes proxies for restoration potential and biodiversity. While IAMs are already in the process to include more and more constraining information to land use allocation, indeed this offline methodology could certainly be helpful. However, it should probably also be discussed or mentioned whether it could be an issue. IAMs may involve many assumptions, but based on these assumptions usually try to consistently follow certain modelling paradigms. Including constraints such as biodiversity as exogenous may create new inconsistencies (breaking some internal logic). The estimates from MPI-ESM on forestation effectivity in terms of net carbon extraction from the atmosphere and cooling potential are rather moderate which may allow more robust conclusions. However, neither globally aggregated area pledges nor socio-economic pathways modeled by IAMs guarantee plausibility or realism. While the authors mention these limitations, their criterion of realism and plausibility and hence consequences for their results remain rather unclear.

L.1: To me it seems like your title neither reflects scope nor methods of your study. I highly recommend to indicate that you work entirely in the modelling world. Neither IAM projections of restoration potential nor climate projections under large-scale forestation can be fully validated against observations.

L.9-10: What about SSP1-2.6? Low warming, high AR (> 800 Mha in 2100) What about LUMIP land use swap between SSP126 and SSP370? Indeed there is some avoided deforestation signal, but still these efforts should not be forgotten.

L.16-17: Linguistic error. Pledges are not a mitigation (technical) but a political tool. Also pledges firstly occur in reality and can only be mimicked in the model world. Be careful not to mix terminology of actual climate history and planning with modeling jargon.

L.31: “overshoot can seem attractive”: to whom (not)? In science vs. in global climate action? Here you could give some context on interest in certain pathways.

L.39: “typically considered as a useful ...” If the picture is not crystal-clear, you may want to balance your introduction with a source or two on contexts and voices not deeming it useful.

L.45: “potential of sequestering...”. I propose to insert “durably” as an important characteristic.

L.45: introduce “carbon (C)”

L.51: I would suggest to use “high estimates” instead of “optimistic”

L.87-90: Contextualize which scenarios in which regions, why. In the IAMs themselves, e.g. SSP1-2.6 by IMAGE has substantial (323 Mha by 2050, 834 Mha by 2100) amounts of AR although this does not directly translate into AR in the ESMs.

L.99: rephrase “unmitigated scenarios”. Scenarios are not mitigated, global warming is.

L.134: What is the reference forest cover data set? How was it generated, with which thresholds of tree cover to qualify forest?

L.140: In my opinion the uncertainty range given should be contextualized with the ensemble size at this point already.

L.146-147: Is the trajectory really Paris-compatible when the time-series of annual global averages needs is smoothed to strictly stay below?

L.182: Could you plot efficiency over time and with uncertainty ranges, e.g. additional to Fig. S3?

L.203-205: From when on could one take a snapshot map (1-year annual mean) of any ensemble member and know whether it is REF or AR?

L.254: Could this be due to MPI-ESM-specific response? Due expect significantly different balances between BGP and BGC responses from other modes?

L.264-266: Well, as IAMs to date fail to include many significant processes, is it really reliable to lean onto their judgement of plausibility? What is your plausibility criterion?

L.270: Great that you mention these limitations. Can you make even clearer what they mean for your results and interpretations?

L.283-284: If these barriers exist, could you at least include some indication (e.g. on the national level) where the cited studies see uncertainty for these reasons, maybe in Fig. 2, 3 and S1? This will enable to reader to judge upon plausibility more on their own.

L.285-287: Could you give some context about the share of needed AR finance organized in reality to date? This will enable to reader to judge upon plausibility more on their own.

L.288: recommendation to insert “in our simulations” between “feature” and “is the early...”

L.316-318: This is a bit short-cut. Could you maybe detail the discussion you provide here spread out through the paragraph for the important studies you consider for comparison.

L.332-342: What about the status of MPI-ESM among other CMIP6-models in terms of cooling behaviour in SSP5-3.4os? What does this mean for the behaviour of AR vs. REF?

L.342: “mitigated emission scenarios” to me seems a bit to far into jargon. Either reorder “under scenarios of strongly mitigated emissions” or rephrase.

L.379: This information on the exact emission scenario came too late for me. I would appreciate some context early on (e.g. in the paragraph l.s 109-118) to contextualize which GMT-trajectory you perturb with AR.

L.383: I understand that the technically easiest adaption of SSP5-3.4os to AR is to deviate from it in 2015. To not have a period until AR launches over which other kinds of LULCC occur. Or for technical simplicity to initiate the simulations in the same year. How does your scenario from 2015 to 2025 differ from observed LULCC? How do you justify the relevance of numeric results like changes in peak GMT, changes in overshoot duration, changes in 2100 GMT (l.s 149-151) when actual AR would occur later just 5 or 10 years?

L.420: You frame interannual variability as noise. But isn't the year-to-year evolution relevant for global scale impacts? What about the comparison of hottest vs. coldest percentiles of the distribution of a running window you just used to eliminate this variability. Have you tried to look at how extremes are changed by AR-induced mechanisms? What are the MPI-ESM-specific signatures, what do you expect to generalize?

L. 436: What is your regridding method? For reproducibility please indicate the algorithm.

L.444-445: Same concerning interpolation: Linearly? Differently? For globally aggregated values or for region-specific ones? Instead, did you consider to interpolate at grid-cell scale once the distribution is completed? How sensitive is your method to such changes?

L.445-446: So, does this mean that you only try to match pledged AR area globally and not regionally? I expect this to make your distribution differ fundamentally from the nationally pledged one. If you can, to defend this doubt, maybe show a plot of how your distribution nationally aggregated compares to national AR pledges. The IAM REMIND-MAgPIE to date already includes LULCC pledges at the national level. So it would be nice to show in the supplement how you do this and to compare to existing methods.

L.451-453: How do you disaggregate/downscale? What are your decision rules and what are they based on? If you use published methods, give references.

L.460-506: Could you maybe provide line plots like in Fig. 3 but showing the sensitivity to different constraints employed? E.g. fully (GRS-ATL-BioDiv) constrained vs. GRS-ATL- constrained vs. only GRS constrained vs. unconstrained?

Fig.1: I think this figure would just need to appear at bigger scaling with more axis details, gridded background and then could serve much more its purpose.

Fig.2: Could you maybe include a layer for regions where literature considers AR potential uncertain beyond quantifiable constraints?

Fig.3: For line plot same as Fig.1. It is important that not only overall upwards vs. downwards trends but e.g. comparison to actually observed trends 2015-2023. can be made. I think including a marker of country pledges in all the region-specific line-plots would help the reader a lot. Also the maps the above-mentioned layer on national-level unquantified constraints (e.g. hatching) could help.

Fig.6: I think the two lines in the center of the plot are not that helpful. They are also the same as in panel of Figure 4, right? I think it could make more sense showing larger maps.

Suppl.Fig.1: Same as Fig. 2 and 3. For all figures representing coarse-scale model data: I would highly appreciate a way of plotting the values as a mesh instead of smooth colour blobs. This alienates the data at smaller scales. Also consider using discrete color-steps in you colormap for clearer comparison of different regions.

Suppl.Fig.2,3: A third / fourth panel on efficiency (removal from atmos / land sequestration) over time in either Fig. 2 or 3 could be very useful for the understanding of the carbon dynamics.

Suppl.Fig.3: I would appreciate the service to the community with a second vertical axis for °C/PgC (b) and PgC /100 Mha (c). Speculatively this could increase the audience size with an intuition for the numbers.

Suppl.Fig.4: What is the added value of the bottom row? Where does it differ from Fig. 6 with more information?

Reviewer #3 (Remarks to the Author):

Response to reviewers

What follows is our detailed point-by-point responses to the reviewers' comments. In several occasions where we consider that some comments are related, these are grouped together, and a single response is provided to avoid repetitions.

Reviewer #1 comments:

Comment:

The authors investigate an ambitious but plausible AR global scenario in line with country pledges. and then assess to what degree a temperature overshoot could be reduced. They develop an AR scenario reaching 595Mha of AR in 2060 and 935 Mha by 2100, based on an anormously large scenarios dbs. They then run fully coupled simulations with the Max Planck Institute's Earth System Model (MPI-ESM). The work as such is impressive, but I believe it is not novel enough for a Nature publication. Biogeochemical cycles or PNAS may be a better suited avenue.

Response:

We thank the reviewer for the overall positive evaluation of our work. We understand that the perception of novelty can be quite subjective and differ among researchers, but we are still confident that our work is novel and can be of significant interest to the scientific community, and is thus suitable for publication in Nature Communications.

In particular, our novelty lies within the following:

- a) There is a lack of Earth system modelling studies applying such ambitious levels of AR under a non-idealized spatiotemporal pattern, even more so with such a setup that:
 - i. technically allows for the full isolation of the effect of AR on the Earth system
 - ii. does not include a high-emissions scenario that implies strong CO₂ fertilization and that bears a possible overestimation of AR mitigation potential
- b) There is a lack of understanding the dynamics of overshoot pathways in the Earth system, which are gaining scientific interest recently due to the increasing policy relevance of overshoot.
- c) The spatiotemporal AR pattern employed here is informed by considering a variety of technoeconomic, societal, and environmental considerations, and a multitude of >1,200 IAM-generated scenarios.
- d) Most studies suffer by the lack of ensemble-members when it comes to variables that normally have strong interannual variability, and robust conclusions cannot be drawn with respect to the signal. As a result, more often than not, robust inference with respect to temperature mitigation cannot be made. Therefore, we overtook the computationally heavy task of creating a very big ensemble-size (20 runs in total) that allows for robust probabilistic treatment and inference.
- e) In the revised version of the manuscript, the applied AR scenario is complemented by a thorough discussion on the associated socioeconomic barriers based on quantified socioeconomic indicators, which is not typical for an Earth system modelling study.

Comment:

As they also show themselves, many of such global scenarios exists and have been assessed in terms of carbon effects. many were also gross overestimates (Griscom et al, Bastin et al..), not taking into account either food production needs or biodiversity effects.

This is now not changed in present study. However they align the results also in terms of reducing overshoot within the possibilities of the Max Planck Model. but e.g. perverse effects in terms of biodiversity are not assessed. they still come with a very high required area 50--900 Mha.

As such a solid study, well written. but I think better suited for journals as mentioned above.

Response:

We thank the reviewer for their comment.

The reviewer argues that food production needs are not taken into account in our study, similar to other gross overestimates. We clarify here that even though we do not directly use food demand and production data to generate

our scenario, such considerations are implicitly included since we are harvesting the Integrated Assessment Model (IAM) database, and the Griscorn restoration map. We comment on that in Lines [403-407]. We refer to the food security concerns in Lines [47 & 443-447].

It should also be noted that we do not claim in our work that biodiversity is not affected by large-scale AR application. Nor do we claim that such ambitious AR does not possibly come with severe consequences, about which stakeholders, policy-makers, and scientists should definitely be informed. In fact we carefully discuss the tradeoffs that such an application can have not only with respect to biodiversity, but also with respect to other societal factors (section “Contextualizing ambitious Afforestation/Reforestation”). In this section, we also very clearly articulate that the present scenario is not meant to serve as a proposed pattern that should be followed in Lines [399-403].

The goal of this study has not been to directly quantify ecological disruption and positive or negative impacts to biodiversity under our AR scenario, as this would require an entire separate work (e.g., Trisos et al., 2020), nor is it claimed in the manuscript that under our AR scenario biodiversity is fully protected. Our declared goal has been reducing the biodiversity impacts of AR application by appropriately prioritizing specific biomes over others within given technoeconomic considerations. To be able to assess the Earth system impacts of ambitious forestation in the range of country pledges, our study augments the existing IAM scenario information by allocating its AR information spatially while avoiding the most severe biodiversity impacts. As we implement AR while preserving croplands globally, such an allocation implies afforestation of (semi-)natural ecosystems and thus destruction of habitats. We thus propose an allocation that reduces the negative impacts in our AR scenario by:

- a) Following restoration potential maps
- b) Excluding natural grasslands and shrublands
- c) Excluding grazing lands that can be considered closer to a pristine state (while acknowledging the possible caveats of using Human Influence maps as a proxy for biodiversity in Lines [641-659])
- d) Replacing grazing land starting from the least biodiverse first

Reviewer #2 comments:

Comment:

This work discusses the effectiveness of forestation in an overshoot scenario, a storyline of future global average surface-near temperature rising above common set targets (e.g., the 1.5°C target in the Paris Agreement) and subsequently falling below, typically within the 21st century. In a single-model ensemble simulation setup, the fully coupled MPI-Earth System Model is used to compute a reference (REF, no forestation) and a forestation (AR) climate scenario. The author’s main claim that such land cover and land use change (LULCC) can mitigate temperature overshoot in the model in terms of height and duration seems to be well-founded regarding presented scenario design and analysis. Another ambition of the study is to apply a “plausible” AR scenario compatible with global scale AR pledges. This is done by combining scenarios from Integrated Assessment Models (IAMs) with restoration potential and biodiversity maps. The study provides useful information on the mitigation potential of AR considering overshoot, however, I have some comments that should be addressed.

Response:

We thank the reviewer for their overall positive evaluation of our work.

Comment:

I would appreciate a more in-depth analysis or discussion of the processes at play. The study mostly offers assessment from the narrow angle of global averages and sums rather than of region-specific regimes of system behaviour. This is incomplete concerning regional effects and extreme climate behaviour, which can be particularly relevant for the biophysical effects. Looking at averages might be enough to judge AR’s overall effectiveness for long-term earth system behaviour, e.g. concerning impacts on the cryosphere, information which partly might be readily available from the performed coupled model simulation. However, a discussion of this relevance also seems to be lacking. If “ambitious”, i.e. global-scale forestation can mitigate modelled temperature overshoot, what does this mean for the oceanic, cryospheric, and land-bound systems? In that sense, what are the significant novel implications for climate science and/or climate action?

Response:

We thank the reviewer for their comment. Following their recommendations, we have significantly extended our analysis to provide a more detailed overview of the Earth system responses under AR.

In particular, we have extended our analysis by including the changes in hydroclimate (Figure R 1), focusing on changes in precipitation, evapotranspiration, cloudiness levels and relative humidity, and discussed the regional patterns identifying hotspots of changes. Overall, AR leads to a wetter hydroclimate over regions of forestation as evident by increases in precipitation, and evapotranspiration, cloudiness, and relative humidity. To provide a more complete picture we also include in our analysis changes in net radiation, latent and sensible heat fluxes, and albedo (Figure R 2). Finally, we further report on changes in sea ice extent globally and over the Arctic, and changes in global average sea surface pH (Figure R 3).

Figure R 1 has now been added as a main figure in our manuscript (Fig. 7). Figure R 2 & 3 have now been added in the Supplementary material (Supplementary Figure 8 & 9). The results and their regional characteristics are presented and discussed in Lines [227-238, 310-319]

Figure R 1: Spatiotemporal pattern of changes in hydroclimate: From left to right, the differences in precipitation (mm/day), evapotranspiration (mm/day), cloud cover fraction, and relative humidity (%) between AR and REF simulations during 2030-2050 (top), the period around peak warming (2050-2070) (mid), and end-of-century (2090-2100) (bottom) are shown. A negative value indicates a reduction in the AR scenario. Dots indicate regions where the difference is statistically insignificant at the 5% level, after correcting for lag-1 temporal autocorrelation (Zwiers & von Storch, 1995).

Figure R 2: Spatiotemporal pattern of changes in radiation and heat fluxes: From left to right, the differences in surface net radiation (W/m^2), latent and sensible heat fluxes (W/m^2), and albedo between AR and REF simulations during 2030-2050 (top), the period around peak warming (2050-2070) (mid), and end-of-century (2090-2100) (bottom) are shown. A negative value indicates a reduction in the AR scenario. Dots indicate regions where the difference is statistically insignificant at the 5% level, after correcting for lag-1 temporal autocorrelation (Zwiers & von Storch, 1995).

Figure R 3: Top row: From left to right, changes in sea ice fraction over the Arctic between AR and REF simulations during 2030-2050, the period around peak warming (2050-2070), and end-of-century (2090-2100) are shown. Bottom row: From left to right, changes in global minimum, average, and maximum sea ice extent (10^7 km^2) for AR (green) and REF (blue) are shown, with the thick lines representing the ensemble mean for each scenario.

Comment:

Another concern related to this is how time of signal emergence, peak temperature difference and interannual variability are treated. Concerning peak global mean 2m T, I understand that a robust comparison needs some smoothing of the interannually fluctuating behaviour of global mean T. However, I think it is important to not overlook differences in extremes induced or enhanced by interannual variability. E.g., how does the running window 10th and the 90th percentile compare between AR and REF? Here I think some work either with statistical or more process analysis is useful to give a less simplified picture of the Earth System behaviour.

L.420: You frame interannual variability as noise. But isn't the year-to-year evolution relevant for global scale impacts? What about the comparison of hottest vs. coldest percentiles of the distribution of a running window you just used to eliminate this variability. Have you tried to look at how extremes are changed by AR-induced mechanisms? What are the MPI-ESM-specific signatures, what do you expect to generalize?

L.146-147: Is the trajectory really Paris-compatible when the time-series of annual global averages needs is smoothed to strictly stay below?

Response:

We thank the reviewer for their comments.

At first, we should clarify that in the old version of the manuscript the time of signal emergence was not estimated by smoothing out interannual variability. Therefore, this has not changed in the revised version.

To address the remaining issues raised and to avoid any biases induced by the user and/or confusion to the reader, the information on peak and end-of-century temperature is now not estimated based on the smoothing and bootstrapping scheme employed in the previous version of the manuscript any more. In the revised version of the manuscript, this temperature information is now rather directly taken from the unsmoothed temperature timeseries of each ensemble member, and uncertainty is expressed in terms of minimum-maximum values obtained from the different members. To robustly capture end-of-century temperature and take into account interannual variability suggesting that the 1.5°C target might be occasionally exceeded, we consider not only 2100, but the pooled data of full 5-year period 2096-2100 across the ensemble members. Similarly, to get robust estimates on peak temperature and avoid biases by likely extremely hot years later or early on, the full 5-year consecutive period yielding the highest average temperature for each ensemble member is considered and the data are then pooled together. We have tested the sensitivity of our results to the chosen period length, as shown in Figure R 4 showing a negligible bias. This is now clarified in the methodology section in Lines [540-545]. Figure R 4 has been added to the Supplementary Material (Supplementary Fig. 11).

Figure R 4: The effect of the choice of period length (x axis) in terms of estimating peak and end-of-century temperature mitigation between AR and REF is shown.

When it comes to estimating the temporal characteristics of the overshoot we choose to keep the smoothing/bootstrapping technique, since a smoothed timeseries allows for a more straightforward identification of overshoot onset and offset timing.

Following the reviewer's suggestion, we have also extended our analysis to different percentiles of temperature both at the gridded level and averaged globally. In particular, in the revised Figure 6 of the manuscript, maps of the 10th (low temperature extreme) and 90th (high temperature extreme) percentiles of temperature at the gridded level are now shown (Figure R 5).

Additionally, we present the 5th, 10th, 25th, 75th, 90th, and 95th percentile of globally averaged daily temperature in Supplementary Figure 3 (Figure R 6). In a given year and for every ensemble member, the Xth percentile of temperature is obtained from the globally averaged daily temperature timeseries. For all these percentiles of globally averaged daily temperature we treat the temperature timeseries with the same statistical treatment presented in the Methods section, to estimate the time of emergence of the signal.

Our results both at the gridded and global level show a robust mitigation of temperature across all percentiles of the temperature distribution, and not only when it comes to the mean. These results are presented and discussed in Lines [162-163, 210-226, 257-259, 320-335].

Figure R 5: Spatiotemporal pattern of temperature change: The in 10th percentile (low temperature extremes, left), mean (mid), and 90th percentile (high temperature extremes, right) of 2m air temperature between AR and REF simulations during 2030-2050 (top), the period around peak warming (2050-2070) (mid), and end-of-century (2090-2100) are shown. A negative difference (blue color) indicates that temperature is lower in the AR scenario. Dots indicate regions where the difference is statistically insignificant at the 5% level, after correcting for lag-1 temporal autocorrelation (Zwiers & von Storch, 1995).

Figure R 6: The panels show different percentiles of 2m air temperature (difference compared to pre-industrial era, expressed here as the 1850-1900 average) for the REF (blue color) and AR (green color) scenarios. The thick lines represent the ensemble mean for each scenario. The bright green shaded region indicates that there is no statistically significant difference between the two scenarios, while the purple shading suggests that a statistically significant difference exists (see Methods). The percentile of temperature used is noted at the title of each panel. In a given year and for every ensemble member, the X^{th} percentile of temperature is obtained from the globally averaged daily temperature timeseries.

Comment:

L.140: In my opinion the uncertainty range given should be contextualized with the ensemble size at this point already.

Response:

Following the reviewer's suggestion, uncertainty range is now contextualized with the ensemble size, by reporting the minimum and maximum values obtained from the different ensemble members.

Comment:

This will also help in the following discussion, what does it mean to find the two scenarios distinguishable? Can you provide examples for contexts in which 0.09°C less globally make a significant difference regionally for certain processes?

Response:

To demonstrate the extent to which reducing the overshoot affects the Earth system, our extended analysis including changes in hydroclimate, energy fluxes, and the ocean (as described above) is presented separately for 2030-2050, the period around peak warming (2050-2070), and the end-of-century (2090-2100) (Figure R 5/ Fig. 6 of the main manuscript).

Comment:

My second comment addresses "plausible" scenario construction. As the authors mention, their judgment of plausibility is based on fitting the AR trajectory to globally aggregated AR pledges and doing so by finding a fitting percentile in AR areal amount within the IAM scenario collection. Additionally, they base the distribution of these area indications across global land on a method that includes proxies for restoration potential and biodiversity. While IAMs are already in the process to include more and more constraining information to land use allocation, indeed this offline methodology could certainly be helpful. However, it should probably also be discussed or mentioned whether it could be an issue. IAMs may involve many assumptions, but based on these assumptions usually try to consistently follow certain modelling paradigms. Including constraints such as biodiversity as exogenous may create new inconsistencies (breaking some internal logic). The estimates from MPI-ESM on forestation effectivity in terms of net carbon extraction from the

atmosphere and cooling potential are rather moderate which may allow more robust conclusions. However, neither globally aggregated area pledges nor socio-economic pathways modeled by IAMs guarantee plausibility or realism. While the authors mention these limitations, their criterion of realism and plausibility and hence consequences for their results remain rather unclear.

L.264-266: Well, as IAMs to date fail to include many significant processes, is it really reliable to lean onto their judgement of plausibility? What is your plausibility criterion?

L.270: Great that you mention these limitations. Can you make even clearer what they mean for your results and interpretations?

Response:

We thank the reviewer for their comment on the plausibility of our scenario.

It should be noted that the usage of the term “spatiotemporally plausible” was meant to highlight the contrast of our scenario with the fully-idealized spatiotemporal patterns employed in many Earth system modelling studies (e.g., cases where AR is applied over all gridcells, or over an entire latitudinal band such as the tropics, and/or all AR is applied at one instance in time etc.), that do not include any technoeconomical, biodiversity, or other consideration across time nor space, and it thus remains unclear to what extent the Earth system responses reported therein remain relevant for real-world applications.

On the contrary, our AR scenario is constrained by a multitude of factors either implicitly - to the extent that they are considered in IAMs - or explicitly - in the form of biodiversity and restoration potential maps. Our scenario could thus resemble future application across space and time to the extent that:

- a) 55% of the forest area increase is reforestation of historically (from 1850 onwards) deforested areas. We argue that one can reasonably assume that a big part of forestation globally in the future would include reforesting previously deforested land mainly because:
 - i. This land can naturally support the growth of forest from a biophysical perspective. Reforestation would thus require less effort and probably accumulate more carbon, thus being more efficient both financially and mitigation-wise.
 - ii. Afforestation would likely jeopardize biodiverse grassland ecosystems which should be protected – as is the case in our study.

We comment on this in Lines [414-416]

- b) The temporal evolution of the global AR target and its disaggregation across the different world economic regions follows technoeconomic constraints.
- c) The global AR target is in the range of country pledges.

“Plausibility” in our case does not suggest that the employed scenario faces no or less hurdles with respect to implementation (e.g., due to issues related to governance, land insecurity, public acceptance etc), that would constitute it “more feasible” than others. In fact - as the reviewer also highlights - we commented on feasibility concerns and discussed such limitations in Lines [51-52, 271-287] of the old manuscript.

In the revised version of the manuscript we also discuss the political feasibility of setting such an ambitious AR target, as reflected by the credibility ratings of the countries undertaking the AR application. In particular, we quantify the percentage of AR across the countries with the higher pledges (Dooley et al., 2022; Self et al., 2023), and also across the countries with the higher AR in our scenario, and demonstrate how all of these countries are countries of either low or very low confidence ratings on their net-zero policies, as estimated by Rogelj et al. (2023). Country-level data are presented in Figure R 7, which has now been added to the Supplementary Material (Supplementary Fig. 1).

The above are discussed in Lines [417-428].

Figure R 7: Country level AR data: The bar plots show country level data of land-based mitigation pledges for AR and ecosystem restoration (Dooley et al. 2022; Self et al. 2023), and the amount of AR achieved by 2060 and 2100 in the scenario employed in this study. The three plots show (top left) the 15 countries with the higher amount of AR reached in 2060 in our scenario, (top right) the 15 countries with the higher AR pledge based on the Land-Gap report estimates (Dooley et al. 2022; Self et al. 2023), and (bottom left) the 15 countries with the higher amount of AR reached in 2100 in our scenario. For these quantities, the cumulative share (%) with increasing number of countries sorted in descending order is shown (bottom right).

Additionally, to allow the readers to judge upon the plausibility of such a scenario on their own, we present a new assessment of the developed AR scenario through the lens of social indicators and the associated tradeoffs (Figure R 8). In particular, we employ: a) a composite indicator of governance, b) an indicator of land tenure insecurity, c) a poverty indicator, d) a map of indigenous and community lands, and e) a map of population density. Detailed information on the data and methodology used is presented in the Supplementary Methods (section “Socioeconomic indicators”). All of these socioeconomic factors are indicative of and associated with an array of possible barriers to implementation of AR, threats to the permanence of newly planted forest, and societal consequences and tradeoffs. We quantify how the developed AR pattern spatially aligns with these indicators (e.g., how much of AR is applied over regions with high poverty) and what that could suggest in terms of possible consequences, barriers to implementation, and threats to permanence. This detailed assessment is presented in Lines [441-469], while Figure R 8 has been added to the Supplementary Material (Supplementary Fig. 10).

Figure R 8: The maps indicate: a) indigenous and community land expressed as a fraction of land per gridcell (Dubertret & Alden Wily, 2015), b) land tenure insecurity expressed as the percentage (%) of people perceiving their land or property to be insecure (Feyertag et al., 2020; Prindex, 2020), c) governance expressed with a composite governance indicator (Kaufmann & Kraay, 2023), d) poverty indicator expressed as the percentage (%) of people below the \$2.15 threshold (World Bank, 2023), and e) population density (people/km²) (WorldPop, 2018). The hatching shows the AR area in the employed scenario. The bar plots show the cumulative AR area (Mha) across different bins of the various socioeconomic indicators.

As the reviewer also acknowledges, in the old version of the manuscript we had clarified that our scenario can be considered “plausible” only to the extent that IAMs can be treated as such, and also openly discussed IAM weaknesses (in Lines [264-270]). However, despite these limitations, the IAMs still remain the main tool at hand when it comes to trying to contextualize the technical, social, and economic developments in the world, and developing constrained spatiotemporal scenarios that go beyond fully-idealized setups. We comment on that in Lines [410-413] of the revised version of the manuscript.

Nevertheless, even though the term “plausibility” - which was used to characterize our scenario - can have a wide variety of meanings (Jewell & Cherp, 2020), we acknowledge that it might possibly be interpreted by some readers as reflecting some underlying normative judgement on feasibility, and thus possibly implying the absence of real-life barriers to financing and implementation of such an ambitious project. Having clarified all that, and despite all limitations have been transparently discussed in detail and expanded in the revised version of the manuscript, to avoid any confusion we now characterize our scenario as “constrained” throughout the text.

To increase visibility, all these issues are now discussed under the new section “Contextualizing ambitious Afforestation/Reforestation”, that follows the Discussion section. As a result, some of the paragraphs of the Discussion section of the old version of the manuscript have been moved to the new section, while the remaining paragraphs have been reordered accordingly to maintain consistency throughout the text.

It should also definitely be acknowledged that pooling multiple AR6-SD scenarios together inevitably breaks the internal consistency found within single IAM scenarios. At the same time, disaggregating the regional AR6-SD information to the gridcell level based on restoration potential and biodiversity maps, as well as not implementing land use transitions other than AR, breaks the internal IAM logic with respect to land use dynamics. Further, selecting an emission trajectory (SSP5-3.4os) independently of the land use pattern employed can also introduce inconsistencies.

For example, achieving net-negative emissions in SSP5-3.4os is heavily based on carbon capture and storage (CCS) (Melnikova et al., 2021), which suggests that additional land pressure would be exerted by the need for large-scale bioenergy CCS application, thus increasing competition with AR over land. It is thus clear that the offline approach employed in our study, introduces some inconsistencies.

However, despite the methodology employed here breaks internal IAM logic, a single IAM scenario has not been used, and a methodology tailored to cover the specific needs of this study has been developed instead. In particular, only by not allowing for land use transitions other than AR to occur, are we able to fully isolate the signal of AR on the Earth system - which is key to our study. This choice inevitably introduces inconsistencies, however it is not arbitrary, but is rather supported by the data shown in Fig. 1 of the main manuscript, that show that in IAMs ambitious AR occurs at the expense of grazing lands, with croplands remaining roughly unchanged. This general IAM logic when it comes to ambitious AR is thus preserved in our scenario. At the same time, to strengthen our approach and preserve the general spatial features of AR reflecting regional-level dynamics and market mechanisms as realized in IAMs, we have additionally utilized regional AR6-SD data as a guide to the spatial disaggregation process. It should also be highlighted that pooling multiple AR6-SD scenarios together eliminates the bias of selecting a single IAM scenario, which would mean selecting a single combination of IAM, SSP, and the configuration and assumptions therein, which would likely also raise questions. It is important to also note that inconsistencies are inevitably introduced even when a single IAM scenario is run with an ESM, due to differences in spatial resolution and land use patterns, and discrepancies in the representation of the biosphere and the carbon cycle (Cheng et al., 2024; Melnikova et al., 2022).

The above are acknowledged and discussed in Lines [385-397].

Comment:

L.1: To me it seems like your title neither reflects scope nor methods of your study. I highly recommend to indicate that you work entirely in the modelling world. Neither IAM projections of restoration potential nor climate projections under large-scale forestation can be fully validated against observations.

Response:

The title of the manuscript has been changed according to the reviewer's recommendations and is now "Temperature overshoot responses to ambitious forestation in an Earth System Model".

Comment:

L.16-17: Linguistic error. Pledges are not a mitigation (technical) but a political tool. Also pledges firstly occur in reality and can only be mimicked in the model world. Be careful not to mix terminology of actual climate history and planning with modeling jargon.

Response:

This has been rephrased in the abstract.

Comment:

L.31: "overshoot can seem attractive": to whom (not)? In science vs. in global climate action? Here you could give some context on interest in certain pathways.

Response:

The whole paragraph has been rephrased and reorganized.

Comment:

L.39: "typically considered as a useful ..." If the picture is not crystal-clear, you may want to balance your introduction with a source or two on contexts and voices not deeming it useful.

Response:

In the old version of the manuscript, following this sentence we mentioned the possible societal, biodiversity, and food security risks in Lines [44-48]. Later on in Lines [51-52] we mentioned that the feasibility and associated socioeconomic

risks of ambitious AR have been questioned. Therefore, to our understanding the old version of the manuscript already offered a quite balanced introduction without trying to present AR as a panacea.

Nevertheless, to avoid misinterpretation we have a) rephrased from “typically considered as a useful” to “can be a useful”, and b) added the concerns over permanence in the list of possible risks in Lines [44-48].

Comment:

L.45: “potential of sequestering...”. I propose to insert “durably” as an important characteristic.

Response:

This has been changed according to the reviewer’s recommendations.

Comment:

L.45: introduce “carbon (C)”

Response:

This has been changed according to the reviewer’s recommendations.

Comment:

L.51: I would suggest to use “high estimates” instead of “optimistic”

Response:

This has been changed according to the reviewer’s recommendations.

Comment:

L.9-10: What about SSP1-2.6? Low warming, high AR (> 800 Mha in 2100) What about LUMIP land use swap between SSP126 and SSP370? Indeed there is some avoided deforestation signal, but still these efforts should not be forgotten.

L.87-90: Contextualize which scenarios in which regions, why. In the IAMs themselves, e.g. SSP1-2.6 by IMAGE has substantial (323 Mha by 2050, 834 Mha by 2100) amounts of AR although this does not directly translate into AR in the ESMs.

Response:

We thank the reviewer for these two comments. We group these and address them together since both refer to AR under SSP1-2.6.

As the reviewer rightly points out, there is a known mismatch between what the marker IAM intended and what eventually was implemented in ESMs. However, based on data available on the SSP database and the AR6 scenario database we have not been able to validate the numbers of AR suggested by the reviewer (Figure R 9) and thus have not included them in the revised version of the manuscript. In particular, based on these data (Figure R 9) the total forest area change is 259 Mha and 516 Mha in 2050 and 2100 respectively. Further, this reflects the net forest area change including both AR and likely deforestation, so maybe to some extent this could justify this discrepancy. However, it is not possible to isolate the amount of AR from neither the databases nor the publication associated with IMAGE-SSP1-2.6 (Doelman et al., 2018).

We would deeply appreciate it if the reviewer could redirect us to these data, if they are available.

Figure R 9: Global forest area under the SSP1-2.6 scenario based on the IMAGE IAM presented in Mha (left axis) and Mha compared to 2010 as a reference (right axis). Data are extracted from AR6 scenario database.

It is due to the above considerations why in the older version of the manuscript we commented that AR is not ambitious enough in LUMIP and other previous studies that were listed, and thus we stated in the abstract that “studies typically apply only moderate levels or idealized AR patterns and under high-emission scenarios”. Nevertheless, we understand the reviewer’s concern that this could possibly be misinterpreted, and therefore: a) we clarify this further in Lines [89-93], and b) we have completely rephrased the abstract to avoid confusion. Results based on the LUMIP study of Loughran et al., (2023) which was published following the initial submission of our manuscript are also presented in Lines [284-288].

Comment:

L.99: rephrase “unmitigated scenarios”. Scenarios are not mitigated, global warming is.

Response:

This has been changed according to the reviewer’s recommendations.

Comment:

L.134: What is the reference forest cover data set? How was it generated, with which thresholds of tree cover to qualify forest?

Response:

We understand that the term “forest cover” might be confusing. We have changed to “tree cover” instead to avoid confusions.

Comment:

L.182: Could you plot efficiency over time and with uncertainty ranges, e.g. additional to Fig. S3?

Response:

The Supplementary Figure (old version: 3, new version: 5) has been changed following the reviewer’s recommendations. In particular, efficiency expressed as atmospheric removal / land sequestration (%) is now plotted. In addition, we have added uncertainty ranges to all quantities in both Supplementary Figures 4 and 5.

Comment:

L.203-205: From when on could one take a snapshot map (1-year annual mean) of any ensemble member and know whether it is REF or AR?

Response:

We thank the reviewer for their comment. To infer upon field significance (i.e. whether the annual average temperature field under AR is statistically significantly different from REF), we first apply at the gridcell level the same test statistic that we applied at globally averaged temperature (see Methods), and then account for the probability of at least one false positive among a multitude of tests (equal to the number of gridcells in our case), called the familywise error rate (Cortés et al., 2020). If a single gridcell is found to be statistically significantly different after accounting for this probability, then we can reject the global null hypothesis that a statistically significant difference between the AR and REF field does not exist.

To increase the robustness of our approach we have performed a sensitivity analysis by: a) pooling temperature data at the gridcell level over a moving window with length ranging from 1 (1-year snapshot) to 10 years, and b) characterizing field significance when the null hypothesis is consistently rejected for a consecutive number of years ranging from 1 to 5, defining as year of emergence the starting year of that period.

We tested the sensitivity of these results by using a variety of commonly used methods including the method by Bonferroni (1936), Walker (1914), Hochberg (1988), Holm (1979), Benjamini and Hochberg (1995), and Benjamini and Yekutieli (2001), yielding similar results.

Results are presented in Figure R 5. Given the highly variable nature of temperature patterns and the ensemble size, field significance at the 5% significance level does not emerge when 1-year snapshots are examined. Field significance however starts emerging as we increase the moving window length (Figure R 10). The results are uncertain for small moving window lengths, but converge to ~2030 as the sample size increases.

This result is now included in the manuscript in Lines [210-212], and the methodology is explained in the Methods section in Lines [556-566]. Figure R 10 has been added to the Supplementary Material (Supplementary Fig. 7)

Figure R 10: Year of emergence of field significance (y axis) using the Bonferroni method (Bonferroni, 1936), depending on the moving window length chosen (x axis), and the number of consecutive years rejecting the null hypothesis needed to declare significance (color). Missing (not plotted) values suggest that a signal has not emerged. Dots are slightly nudged across the vertical axis to aid interpretation.

Comment:

L.254: Could this be due to MPI-ESM-specific response? Due expect significantly different balances between BGP and BGC responses from other modes?

Response:

To our knowledge, there is no study that has robustly addressed this research question. For example, based on the LUMIP esm-ssp585-ssp126Lu simulations Loughran et al. (2023) reported no significant temperature mitigation globally (Lines [284-288]). As a result, the biogeochemical cooling therein is too weak to offer any insight to the relative strength of biogeophysically-induced warming and biogeochemical cooling at the local level.

Therefore, the extent to which this model behavior can be consistent across models remains to be assessed. This is further clarified in Line [335].

Comment:

L.283-284: If these barriers exist, could you at least include some indication (e.g. on the national level) where the cited studies see uncertainty for these reasons, maybe in Fig. 2, 3 and S1? This will enable to reader to judge upon plausibility more on their own.

Response:

In the updated version of the manuscript, a more detailed quantified post-assessment of our scenario based on several socioeconomic factors including land tenure insecurity is discussed in the new section “Contextualizing ambitious Afforestation/Reforestation” and is presented in Supplementary Fig. 10 (as described in our responses above and shown in Figure R 8).

Comment:

L.285-287: Could you give some context about the share of needed AR finance organized in reality to date? This will enable to reader to judge upon plausibility more on their own.

Response:

As the reviewer rightly suggests, until this day lack of both public and private finance has been one of the key barriers to meeting global restoration needs, and scaling-up funding would be a big challenge that would require the implementation of new green finance mechanisms and regulations and subsidies for restoration (Löfqvist et al., 2023). This is now discussed in Lines [438-440].

Comment:

L.288: recommendation to insert “in our simulations” between “feature” and “is the early...”

Response:

This has been changed according to the reviewer’s recommendations.

Comment:

L.316-318: This is a bit short-cut. Could you maybe detail the discussion you provide here spread out through the paragraph for the important studies you consider for comparison.

Response:

This paragraph has now been refined and we comment on each of the selected studies in more detail.

Comment:

L.332-342: What about the status of MPI-ESM among other CMIP6-models in terms of cooling behaviour in SSP5-3.4os? What does this mean for the behaviour of AR vs. REF?

Response:

We thank the reviewer for their comment.

Both peak and end-of-century temperature under REF are at the lower range of the CMIP6 SSP5-3.4os multi-model ranges (not including MPI-ESM) of 2-4.35°C and 1.39-3.47°C respectively, as recently reported by Asaadi et al. (2024), suggesting that MPI-ESM is among the models with the stronger cooling behavior as a response to negative emissions. However, direct comparisons of the results reported here with those of Asaadi et al. (2024) cannot be made with confidence, since in our study land-use under REF is constant, even though emissions follows the emission trajectory of SSP5-3.4os. It should also be noted that the ranges reported therein are not based on multiple ensemble members.

Despite the sensitivity of MPI-ESM to negative emissions not having been directly compared against other models (Asaadi et al., 2024; Melnikova et al., 2021) and uncertainty remaining, it is the additional sequestration under AR and

the consequent feedbacks that determine temperature mitigation compared to REF. Therefore, it is rather the feedback strength within MPI-ESM that would play the key role.

This is now discussed in Lines [289-309]

Comment:

L.342: "mitigated emission scenarios" to me seems a bit to far into jargon. Either reorder "under scenarios of strongly mitigated emissions" or rephrase.

Response:

This has been changed according to the reviewer's recommendations.

Comment:

L.379: This information on the exact emission scenario came too late for me. I would appreciate some context early on (e.g. in the paragraph l.s 109-118) to contextualize which GMT-trajectory you perturb with AR.

Response:

This has been added in the Introduction (Lines [118-120]).

Comment:

L.383: I understand that the technically easiest adaption of SSP5-3.4os to AR is to deviate from it in 2015. To not have a period until AR launches over which other kinds of LULCC occur. Or for technical simplicity to initiate the simulations in the same year. How does your scenario from 2015 to 2025 differ from observed LULCC? How do you justify the relevance of numeric results like changes in peak GMT, changes in overshoot duration, changes in 2100 GMT (l.s 149-151) when actual AR would occur later just 5 or 10 years?

Response:

We thank the reviewer for their comment. As they rightly assume, branching off from 2015 is the technically easiest way to adapt SSP5-3.4os. According to FAO data, forest area has decreased by 29.96 Mha from 2015 to 2021. By that time, AR amounts to ~5 Mha, which corresponds to 0.5% of total AR application in our scenario (935 Mha), thus yielding a net difference of ~35 Mha of forest area with historical data. Most importantly however, by 2021 land C stocks are increased by 0.81 PgC in the AR compared to the REF scenario, constituting 0.78% of total additional sequestration.

These numbers demonstrate that the bulk of forest area increase and additional C sequestration under AR occur after 2025 in our simulations, which suggests that our results remain relevant with respect to overshoot dynamics and mitigation, regardless of this discrepancy with recent historical data.

We discuss the above in Lines [669-677].

Comment:

L.445-446: So, does this mean that you only try to match pledged AR area globally and not regionally? I expect this to make your distribution differ fundamentally from the nationally pledged one. If you can, to defend this doubt, maybe show a plot of how your distribution nationally aggregated compares to national AR pledges. The IAM REMIND-MAGPIE to date already includes LULCC pledges at the national level. So it would be nice to show in the supplement how you do this and to compare to existing methods.

Response:

At the time of writing of the initial manuscript version, a complete list of national-level data and the corresponding land requirements were not available in the Land-Gap report, or elsewhere to our knowledge. As a result, only globally and not regionally/nationally pledged area was considered, as the reviewer rightly points out. The fact that the pledges are considered at the global level is now clarified in multiple occasions throughout the text.

Only after the submission of our manuscript, national AR country pledges have been included in an update of the Land-Gap report (Self et al., 2023). It should be noted that in this recent update of the Land-Gap report total country pledges

by 2060 amount up to 490 Mha (Self et al., 2023), which is 22.5% less compared to the previous estimate of 633 Mha (Dooley et al., 2022). This new estimate is now clearly reported throughout all text and figures.

However, there are still not available estimates for all countries, especially when it comes to Latin America, Africa, and Asia (Dooley et al., 2022; Self et al., 2023), which would force us to not apply AR over these countries, if gridcell level allocation were to be consistent with pledges at the country-level. More importantly however, it should be noted that the pledge from Saudi Arabia accounts for 42% of the total AR target globally, and can only be met through international offsets in addition to domestic land-based CDR (Self et al., 2023). Since it is unclear where this AR will be applied, this uncertainty suggests that if AR in the range of country pledges is to be applied, then our spatial pattern should not strictly be defined by the narrow region/country boundaries, and therefore the disaggregation of the global target has been based on regional AR6-SD estimates. At the same time, employing the AR6-SD allows for an explicit treatment of the temporal evolution of AR across the century based on technoeconomic considerations, instead of simply interpolating pledges up to 2060, and arbitrarily extrapolating thereafter.

We comment on this in Lines [594-602].

Nevertheless, we still show a comparison of country-level AR in our scenario with the data from Land-Gap Report 2023 update in Supplementary Figure 1 (Figure R 7), and show and discuss the characteristics of country pledges, and how they compare to the country-level AR in our scenario in Lines [134-136,419-426].

Comment:

L.451-453: How do you disaggregate/downscale? What are your decision rules and what are they based on? If you use published methods, give references.

Response:

We thank the reviewer for their comment. This was indeed not clarified in the older version of the manuscript.

We treat regional data in a similar way as the global-level data. For every region we pool available scenarios together, and select the 90th percentile of the pooled regional yearly forest area change. Given that a different number of scenarios is available at the regional (1,124) compared to the global level (1,259), and that we do not select a single scenario, but rather choose a percentile, the sum of all regional 90th percentiles might not be equal to the 90th percentile of global estimates. As a result, the regional estimates are rescaled so that their sum matches the yearly global target (preserving their relative magnitude with respect to their sum).

This is now clarified in Lines [606-612].

Comment:

L. 436: What is your regriding method? For reproducibility please indicate the algorithm.

Response:

The use of the conservative remapping algorithm is now mentioned in the text.

Comment:

L.460-506: Could you maybe provide line plots like in Fig. 3 but showing the sensitivity to different constraints employed? E.g. fully (GRS-ATL-BioDiv) constrained vs. GRS-ATL- constrained vs. only GRS constrained vs. unconstrained?

Response:

To test the sensitivity of the AR scenario development algorithm to the different constraints employed, we re-run the algorithm with different configurations. In particular, we develop scenarios where: a) only GRS and ATL restoration maps (i.e., no explicit biodiversity consideration), b) only GRS, and biodiversity maps c) no restoration or biodiversity maps are used to constrain the scenario development. Results are shown in Figure R 11.

Even though differences arise between the configurations, the patterns share some similarity with the AR scenario employed in this study. It should be noted that pastures are an integral part of both GRS and ATL (the first being a subset of the latter). As a result, prioritizing pastures can partly compensate for not using restoration potential maps

to guide AR, due to the inherent consistency between the two. At the same time, consideration of biodiversity does not affect the partitioning between rangelands and pastures that are given up, since biodiversity is only considered for rangelands. Therefore, since pastures are generally given up first in our algorithm regardless of the configuration tested, the rough AR pattern is not heavily sensitive to the different constraints employed. However, considering biodiversity affects the spatial pattern of AR mostly since it determines the regions where rangelands can be considered closer to a pristine state, and thus excluded from AR. Even though the AR pattern by 2100 unavoidably converges to the available grazing land, the differences across time can be more pronounced as the partitioning between pastures and rangelands, and the specifics of each configuration change.

This sensitivity analysis is now presented in the Supplementary Material under the section “Sensitivity of AR scenario development”, while Figure R 11 has also been added in the Supplementary Material (Supplementary Figure 12).

Comment:

L.444-445: Same concerning interpolation: Linearly? Differently? For globally aggregated values or for region-specific ones? Instead, did you consider to interpolate at grid-cell scale once the distribution is completed? How sensitive is your method to such changes?

Response:

Interpolation of the 5-year AR6-SD data to yearly both at the global and regional level is performed with a piecewise cubic hermite interpolating polynomial algorithm. This is now clarified in Lines [611-612].

To test the sensitivity of our method to the temporal interpolation, we have developed a scenario where the 5-year IAM data are not interpolated at the yearly level. Instead, the cumulative 5-year AR target is rather directly distributed with the algorithm at the gridcell-level every five years. After that, the AR within the gridcell can be temporally interpolated (e.g., linearly). The results presented in Figure R 12 show that not interpolating the 5-year IAM values yields little differences in the AR pattern compared to the employed AR scenario both at 2070 and 2100. These differences can be explained by the 10% threshold posed to the yearly forest fraction increase over a gridcell, which becomes more important when the full 5-year AR target is distributed at once by the algorithm. However, when the maximum yearly cover fraction increase over a gridcell is increased to 30%, the differences to the AR scenario are practically eliminated.

We refrained from including this sensitivity analysis in the Supplementary Material to keep the material concise.

Figure R 11: The sensitivity of the AR scenario development algorithm to the different constraints employed is shown for configurations where the scenario is: a) constrained only by GRS and ATL (GRS+ATL), b) constrained only by GRS and biodiversity (GRS+BIO), and c) unconstrained. Left: Line plots show the changes in pasture (top) and rangeland (bottom) for the different scenarios, where also the scenario employed in this study is shown (noted as "AR scenario"). Maps: The maps show the difference in forest fraction between the different configurations tested here and the AR scenario employed in this study in 2070 and 2100.

Figure R 12: The sensitivity of the AR scenario development algorithm to the temporal interpolation is shown for configurations where: a) 5-year IAM data are not interpolated at the yearly level, but instead the cumulative 5-year AR target is rather directly distributed with the algorithm at the gridcell-level every five years (“Not interpolated”), and b) same as (a), but with increasing the maximum yearly increase in cover fraction allowed from 10% to 30% (“Not interpolated increased yearly capacity”). Left: Line plots show the changes in pasture (top) and rangeland (bottom) for the different scenarios, where also the scenario employed in this study is shown (noted as “AR scenario”). Maps: The maps show the difference in forest fraction between the different configurations tested here and the AR scenario employed in this study in 2070 and 2100.

Comment:

Fig.1: I think this figure would just need to appear at bigger scaling with more axis details, gridded background and then could serve much more its purpose.

Response:

The figure has been refined.

Comment:

Fig.2: Could you maybe include a layer for regions where literature considers AR potential uncertain beyond quantifiable constraints?

Response:

Spatial information on various socioeconomic indicators is now presented in Supplementary Fig. 10 (Figure R 7, and as described in our responses above), with the AR spatial pattern overlaid with hatching to ease interpretation. Aggregate AR values across various ranges of the socioeconomic indicators are also shown with bar plots and discussed in Lines [447- 469].

Comment:

Fig.3: For line plot same as Fig.1. It is important that not only overall upwards vs. downwards trends but e.g. comparison to actually observed trends 2015-2023. can be made. I think including a marker of country pledges in all the region-specific line-plots would help the reader a lot. Also the maps the above-mentioned layer on national-level unquantified constraints (e.g. hatching) could help.

Response:

We comment on the observed trends in Lines [669- 677]. Given the substantially lower amounts, we refrain from plotting observed trend, to not make the figure overwhelmed with subsidiary information.

Comment:

Fig.6: I think the two lines in the center of the plot are not that helpful. They are also the same as in panel of Figure 4, right? I think it could make more sense showing larger maps.

Response:

Fig. 6 has now been revised, the lines have been removed, and now the mean, 10th (low temperature extreme) and 90th (high temperature extreme) percentiles of temperature for 2030-2050, 2050-2070 (period around peak warming), and 2090-2100 (end of century) are shown.

Comment:

Suppl.Fig.1: Same as Fig. 2 and 3. For all figures representing coarse-scale model data: I would highly appreciate a way of plotting the values as a mesh instead of smooth colour blobs. This alienates the data at smaller scales. Also consider using discrete color-steps in you colormap for clearer comparison of different regions.

Response:

Across all figures containing maps, the style of plotting has now changed to a mesh, and the colormaps used are now more discrete.

Comment:

Suppl.Fig.2,3: A third / fourth panel on efficiency (removal from atmos / land sequestration) over time in either Fig. 2 or 3 could be very useful for the understanding of the carbon dynamics.

Response:

Efficiency expressed as “atmospheric removal / land sequestration” has been added, following the reviewer’s recommendations.

Comment:

Suppl.Fig.3: I would appreciate the service to the community with a second vertical axis for °C/PgC (b) and PgC /100 Mha (c). Speculatively this could increase the audience size with an intuition for the numbers.

Response:

This has been changed according to the reviewer’s recommendations.

Comment:

Suppl.Fig.4: What is the added value of the bottom row? Where does it differ from Fig. 6 with more information?

Response:

The reviewer is right in their remark that bottom-row maps presented the same information as Fig. 6, and were only presented to provide context for the upper- and middle- row maps of the Supplementary Figure, without one needing to consult with Fig. 6 at the same time. We understand that this might cause confusion, so this has now been removed.

Reviewer #3 comments:

Comment:

Response:

We thank the Early Career Researcher for their contribution to this review.

Referenes

- Asaadi, A., Schwinger, J., Lee, H., Tjiputra, J., Arora, V., Séférian, R., Liddicoat, S., Hajima, T., Santana-Falcón, Y., & Jones, C. D. (2024). Carbon cycle feedbacks in an idealized simulation and a scenario simulation of negative emissions in CMIP6 Earth system models. *Biogeosciences*, *21*(2), 411–435. <https://doi.org/10.5194/bg-21-411-2024>
- Benjamini, Y., & Hochberg, Y. (1995). Controlling the False Discovery Rate: A Practical and Powerful Approach to Multiple Testing. *Journal of the Royal Statistical Society: Series B (Methodological)*, *57*(1), 289–300. <https://doi.org/10.1111/j.2517-6161.1995.tb02031.x>
- Benjamini, Y., & Yekutieli, D. (2001). The control of the false discovery rate in multiple testing under dependency. *The Annals of Statistics*, *29*(4), 1165–1188. <https://doi.org/10.1214/aos/1013699998>
- Bonferroni, C. (1936). Teoria statistica delle classi e calcolo delle probabilita. *Pubblicazioni Del R Istituto Superiore Di Scienze Economiche e Commerciali Di Firenze*, *8*, 3–62.
- Cortés, J., Mahecha, M., Reichstein, M., & Brenning, A. (2020). Accounting for multiple testing in the analysis of spatio-temporal environmental data. *Environmental and Ecological Statistics*, *27*(2), 293–318. <https://doi.org/10.1007/s10651-020-00446-4>
- Doelman, J. C., Stehfest, E., Tabeau, A., van Meijl, H., Lassaletta, L., Gernaat, D. E. H. J., Hermans, K., Harmsen, M., Daioglou, V., Biemans, H., van der Sluis, S., & van Vuuren, D. P. (2018). Exploring SSP land-use dynamics using the IMAGE model: Regional and gridded scenarios of land-use change and land-based climate change mitigation. *Global Environmental Change*, *48*, 119–135. <https://doi.org/10.1016/j.gloenvcha.2017.11.014>
- Dooley, K., Keith, H., Larson, A., Catacora-Vargas, G., Carton, W., Christiansen, K., Baa, O., Frechette, A., Hugh, S., Ivetic, N., Ching, L., Lund, J., Luqman, M., Mackey, B., Monterroso, I., Ojha, H., Perfecto, I., Riamit, K., Robiou du Pont, Y., & Vargas Sáenz, A. (2022). *The Land Gap Report 2022*. <https://www.landgap.org/>
- Dubertret, F., & Alden Wily, L. (2015). *Percent of Indigenous and Community Lands. Data file from LandMark: The Global Platform of Indigenous and Community Lands. Available at: Wwww.landmarkmap.org.* [dataset].
- Feyertag, J., Childress, M., Flynn, R., Langdown, I., Locke, A., & Nizalov, D. (2020). *Prindex comparative report: A global assessment of perceived tenure security from 140 countries.* Prindex. <https://www.prindex.net/reports/prindex-comparative-report-july-2020/>
- Hochberg, Y. (1988). A sharper Bonferroni procedure for multiple tests of significance. *Biometrika*, *75*(4), 800–802. <https://doi.org/10.1093/biomet/75.4.800>

- Holm, S. (1979). A Simple Sequentially Rejective Multiple Test Procedure. *Scandinavian Journal of Statistics*, 6(2), 65–70.
- Jewell, J., & Cherp, A. (2020). On the political feasibility of climate change mitigation pathways: Is it too late to keep warming below 1.5°C? *WIREs Climate Change*, 11(1), e621. <https://doi.org/10.1002/wcc.621>
- Kaufmann, D., & Kraay, A. (2023). *Worldwide Governance Indicators, 2023 Update* (www.govindicators.org) [dataset]. <https://www.worldbank.org/en/publication/worldwide-governance-indicators>
- Löfqvist, S., Garrett, R. D., & Ghazoul, J. (2023). Incentives and barriers to private finance for forest and landscape restoration. *Nature Ecology & Evolution*, 7(5), 707–715. <https://doi.org/10.1038/s41559-023-02037-5>
- Loughran, T. F., Ziehn, T., Law, R., Canadell, J. G., Pongratz, J., Liddicoat, S., Hajima, T., Ito, A., Lawrence, D. M., & Arora, V. K. (2023). Limited Mitigation Potential of Forestation Under a High Emissions Scenario: Results From Multi-Model and Single Model Ensembles. *Journal of Geophysical Research: Biogeosciences*, 128(12), e2023JG007605. <https://doi.org/10.1029/2023JG007605>
- Melnikova, I., Boucher, O., Cadule, P., Ciais, P., Gasser, T., Quilcaille, Y., Shiogama, H., Tachiiri, K., Yokohata, T., & Tanaka, K. (2021). Carbon Cycle Response to Temperature Overshoot Beyond 2°C: An Analysis of CMIP6 Models. *Earth's Future*, 9(5), e2020EF001967. <https://doi.org/10.1029/2020EF001967>
- Prindex. (2020). *Prindex 2020 global dataset* [dataset]. <https://www.prindex.net/data/>
- Rogelj, J., Fransen, T., den Elzen, M. G. J., Lamboll, R. D., Schumer, C., Kuramochi, T., Hans, F., Mooldijk, S., & Portugal-Pereira, J. (2023). Credibility gap in net-zero climate targets leaves world at high risk. *Science*, 380(6649), 1014–1016. <https://doi.org/10.1126/science.adg6248>
- Self, A., Burdon, R., Lewis, J., Riggs, P., & Dooley, K. (2023). *The Land Gap Report: 2023 Update*. <https://www.landgap.org/>
- Trisos, C. H., Merow, C., & Pigot, A. L. (2020). The projected timing of abrupt ecological disruption from climate change. *Nature*, 580(7804), 496–501. <https://doi.org/10.1038/s41586-020-2189-9>
- Walker, G. T. (1914). *Correlation in Seasonal Variations of Weather, III: On the Criterion for the Reality of Relationships Or Periodicities*. Meteorological Office.
- World Bank. (2023). *Geospatial Poverty Portal*. World Bank Group [dataset]. <https://pip.worldbank.org/home>
- WorldPop. (2018). *WorldPop* (www.worldpop.org—School of Geography and Environmental Science, University of Southampton; Department of Geography and Geosciences, University of Louisville; Departement de

Geographie, Universite de Namur) and Center for International Earth Science Information Network (CIESIN), Columbia University) [dataset]. www.worldpop.org

Zwiers, F. W., & von Storch, H. (1995). Taking Serial Correlation into Account in Tests of the Mean. *Journal of Climate*, 8(2), 336–351. [https://doi.org/10.1175/1520-0442\(1995\)008<0336:TSCIAI>2.0.CO;2](https://doi.org/10.1175/1520-0442(1995)008<0336:TSCIAI>2.0.CO;2)

REVIEWERS' COMMENTS

Reviewer #2 (Remarks to the Author):

Summary and comments:

This work, an update to “Ambitious forestation can mitigate temperature overshoot”, shows significant improvement, in particular a broader analysis, a deeper evaluation of the forcing scenario, and more careful reflections and discussions on the method and results. These changes address not only the concerns raised by me, but - it seems like - also the ones brought up by the first referee. Now, the manuscript presents nicely the novelty of this work, gives an outstanding introduction and contextualizes the findings in a useful manner within a separate section. I recommend the manuscript for publication after some minor adjustments.

Line-by-line comments:

Abstract: The title has been adapted to make clear that the results are based on one ESM. In the abstract this should be concretized. I suggest to modify the sentence (line 14-15) to make clear that the results are based on the MPI model (by explicitly mentioning the name of the model).

Introduction: This is a very good introduction now. Many aspects introduced. Outstanding compilation.

L. 20: I think it should say “net-zero greenhouse gas emissions”

L.123: Here at the end of the introduction, mentioning your section 4 could help the reader.

L.305-307: Here, not only TCR but also ZEC and findings from the flat10 experiments could give helpful context on how MPI-ESM behaves within an ensemble of ESMs.

Conclusion: Informative paragraph. Either in the discussion or here, I recommend to collect and summarize dispersedly mentioned efforts needed beyond this study to further evaluate the effectiveness of forestation as strategy to contribute to mitigate global warming.

L. 573-575: I would remove the word “robustly” and mention explicitly the model used. For example: “Our results based on MPI-ESM ...”

L 576-578: The biochemical cooling is contrasted against “local warming”. There should probably be a mentioning of biophysical effects. Otherwise it is contrasting “biochemical” against “local”. One of them referring to processes and the other to a spatial scale. In addition, the sentence seems to be slightly overstated considering that there still seem to be significantly warmer grid cells in

Figure 6.

Reviewer #3 (Remarks to the Author):

Response to reviewers

What follows is our detailed point-by-point responses to the reviewers' comments.

Reviewer #2 comments:

Comment:

Summary and comments:

This work, an update to "Ambitious forestation can mitigate temperature overshoot", shows significant improvement, in particular a broader analysis, a deeper evaluation of the forcing scenario, and more careful reflections and discussions on the method and results. These changes address not only the concerns raised by me, but - it seems like - also the ones brought up by the first referee. Now, the manuscript presents nicely the novelty of this work, gives an outstanding introduction and contextualizes the findings in a useful manner within a separate section. I recommend the manuscript for publication after some minor adjustments.

Introduction: This is a very good introduction now. Many aspects introduced. Outstanding compilation.

Response:

We thank the reviewer for their overall positive evaluation of the revised version of our manuscript.

Comment:

Line-by-line comments:

Abstract: The title has been adapted to make clear that the results are based on one ESM. In the abstract this should be concretized. I suggest to modify the sentence (line 14-15) to make clear that the results are based on the MPI model (by explicitly mentioning the name of the model).

Response:

This has been changed according to the reviewer's recommendations.

Comment:

L. 20: I think it should say "net-zero greenhouse gas emissions"

Response:

This has been corrected.

Comment:

L.123: Here at the end of the introduction, mentioning your section 4 could help the reader.

Response:

The discussion on socioeconomic tradeoffs is now mentioned in the introduction (Lines 113-114).

Comment:

L.305-307: Here, not only TCR but also ZEC and findings from the flat10 experiments could give helpful context on how MPI-ESM behaves within an ensemble of ESMs.

Response:

Following the reviewers' suggestions, we now comment on the behaviour of MPI-ESM compared to other models based on the ZECMIP simulations (Lines 283-285).

Comment:

Conclusion: Informative paragraph. Either in the discussion or here, I recommend to collect and summarize dispersedly mentioned efforts needed beyond this study to further evaluate the effectiveness of forestation as strategy to contribute to mitigate global warming.

Response:

Changes have been made in the Conclusion according to the reviewer's recommendations (Lines 453-462).

Comment:

L. 5473-5475: I would remove the word "robustly" and mention explicitly the model used. For example: "Our results based on MPI-ESM ..."

Response:

This has been changed according to the reviewer's recommendations (Line 448).

Comment:

L 5476-5478: The biochemical cooling is contrasted against "local warming". There should probably be a mentioning of biophysical effects. Otherwise it is contrasting "biochemical" against "local". One of them referring to processes and the other to a spatial scale. In addition, the sentence seems to be slightly overstated considering that there still seem to be significantly warmer grid cells in Figure 6.

Response:

This sentence has been clarified (Lines 450-453).

Reviewer #3 comments:

Response:

We thank the Early Career Researcher for their contribution to this review.